# Emergence of the Affect from the Variation in the Whole-Brain Flow of Information

**DOI:** 10.3390/brainsci10010008

**Published:** 2019-12-21

**Authors:** Soheil Keshmiri, Masahiro Shiomi, Hiroshi Ishiguro

**Affiliations:** 1Advanced Telecommunications Research Institute International (ATR), Kyoto 619-0237, Japan; m-shiomi@atr.jp (M.S.); ishiguro@sys.es.osaka-u.ac.jp (H.I.); 2Graduate School of Engineering Science, Osaka University, Osaka 565-0871, Japan

**Keywords:** Granger causality, functional connectivity, information flow, affect, brain signal variability

## Abstract

Over the past few decades, the quest for discovering the brain substrates of the affect to understand the underlying neural basis of the human’s emotions has resulted in substantial and yet contrasting results. Whereas some point at distinct and independent brain systems for the Positive and Negative affects, others propose the presence of flexible brain regions. In this respect, there are two factors that are common among these previous studies. First, they all focused on the change in brain activation, thereby neglecting the findings that indicate that the stimuli with equivalent sensory and behavioral processing demands may not necessarily result in differential brain activation. Second, they did not take into consideration the brain regional interactivity and the findings that identify that the signals from individual cortical neurons are shared across multiple areas and thus concurrently contribute to multiple functional pathways. To address these limitations, we performed Granger causal analysis on the electroencephalography (EEG) recordings of the human subjects who watched movie clips that elicited Negative, Neutral, and Positive affects. This allowed us to look beyond the brain regional activation in isolation to investigate whether the brain regional interactivity can provide further insights for understanding the neural substrates of the affect. Our results indicated that the differential affect states emerged from subtle variation in information flow of the brain cortical regions that were in both hemispheres. They also showed that these regions that were rather common between affect states than distinct to a specific affect were characterized with both short- as well as long-range information flow. This provided evidence for the presence of simultaneous integration and differentiation in the brain functioning that leads to the emergence of different affects. These results are in line with the findings on the presence of intrinsic large-scale interacting brain networks that underlie the production of psychological events. These findings can help advance our understanding of the neural basis of the human’s emotions by identifying the signatures of differential affect in subtle variation that occurs in the whole-brain cortical flow of information.

## 1. Introduction

In psychology, the term “affect” refers to anything that is emotional [1]. It forms the individuals’ basic ability to experience pleasant/unpleasant feelings to express these subjective mental states in terms of such attributes as positive or negative [2]. This ability is considered to be a fundamental property of individuals’ emotion [3,4,5]. It begins from the early days in individuals’ lives [6,7] and appears to form the unifying and common concept across cultures [8,9].

The quest for discovering the brain substrates of the affect has witnessed a growing surge in recent decades. These efforts are broadly identified by two mainstream approaches [1]. They are the locationist and the psychological constructionist approaches. The locationist approach [10,11] hypothesizes that a single brain region consistently and specifically activates across instances of a single emotion category/state (e.g., Positive state). It also argues that such states are biologically basic (i.e., they cannot be broken down into more basic psychological components) [12,13,14]. On the other hand, the psychological constructionist approach [15,16,17] states that the same brain areas consistently activate across the instances of a range of emotion categories. In other words, this viewpoint maintains that no brain region is specifically dedicated to any particular emotion category [1]. Lindquist et al. [2] further categorize these approaches under three major hypotheses: the bipolarity hypothesis [18], the bivalent hypothesis [19,20,21], and the affective workspace hypothesis [22]. The bipolarity hypothesis [18] considers the Positive and Negative affect to form the opposite ends of a single dimension [23,24]. On the other hand, the bivalent hypothesis [19,20,21] emphasizes on the presence of two distinct and independent brain systems for the Positive and Negative affects [19,20,21]. Contrary to these two hypotheses, the affective workspace hypothesis [22] argues that the Positive and Negative affect are the brain states that are supported by flexible rather than a consistently specific set of brain regions [25]. Such a divide in neural substrates of the affect is further escalated by the results of meta-analyses that provide contrasting yet compelling evidence for and against these hypotheses [2,26,27,28,29,30]. For instance, whereas Vytal and Hamann [26] credited the bipolarity hypothesis, Lindquist et al. [2] provided stronger support for the affective workspace hypothesis.

Our overview of the literature on the brain substrates of the affect identifies two factors that play pivotal roles in discrepancies among their findings. First, it identifies that these studies primarily focused on the change in brain activation to call a specific [31,32,33,34] or subset [27] of the brain regions responsible for experiencing an affect. However, this approach neglects the findings that indicate the brain activation and its information content do not necessarily modulate [35]. It also does not take into account that the stimuli with equivalent sensory and behavioral processing demands may not necessarily result in differential brain activation [36]. Second, it indicates that the previous studies did not consider the crucial role of functional interactivity between the brain regions [37]. As a result, they did not take into account the fact that signals from individual cortical neurons are shared across multiple areas and thus concurrently contribute to multiple functional pathways [38].

In this regard, the nonlinear dynamical system analysis [39,40] frames the study of the brain functioning in terms of the interaction between its regions. Specifically, it treats the brain as a complex system [41,42] whose dynamics and ongoing activity [43] orchestrates its cognitive functions [44,45,46,47]. In this respect, the Granger causality (G-causality) [48,49] has found widespread use in neuroscience [50,51,52]. The G-causality is based on predictability and precedence among two or more events that occur at the same time. In the language of G-causality, a variable *X* is said to G-cause a variable *Y* if the past of *X* contains information that helps predict the future of *Y* over and above information already in the past of *Y*. Although the Granger causality is based on linear vector autoregressive (VAR) [48,49,51] and hence linear in nature, it approximates [53,54] the transfer entropy [55] i.e., a nonlinear directional measure of mutual information. An advantage of using Granger causality is that, unlike the transfer entropy’s complicated estimation [55,56,57], its well-established mathematical formulation and known statistical properties allow for straightforward tests of significance [58,59]. Over the past, there has been concern for the use of Granger causality in the neuroscience [60,61]. However, a number of subsequent analyses have provided evidence for its utility in such analyses [62,63,64].

In this article, we utilize the Granger causality to investigate the whole-brain functional interactivity in terms of information flow between different brain regions in response to different affects. We achieve this objective by utilizing the Shanghai Jiao Tong University (SJTU) Emotion EEG Dataset (SEED) [65] that is a collection of human subjects’ sixty-two-channel EEG recordings. EEG is an electrophysiological monitoring method that records the brain’s spontaneous electrical activity over a period of time. These electrical activities are due to the voltage fluctuations induced by ionic current within the neurons [66]. EEG recordings are generally acquired through multiple electrodes. In a SEED experiment, the sixty-two-channel EEG recordings of the participants took place while they watched fifteen movie clips (four minutes in duration) whose contents elicited three distinct affect: Negative, Neutral, and Positive.

Although previous research aimed at identifying the functional interaction among brain regions, it mostly framed such an interactivity in terms of statistical association (e.g., correlation) [67]. However, this approach to the study of brain cortical regional interactivity is problematic since such associations as correlation can arise in a variety of ways that do not entail causal relation (i.e., directional flow of information) [67]. As a result, they do not allow for understanding the mapping between such associations and their underlying neural substrates [67,68]. Addressing these shortcomings can be facilitated by utilization of such approaches as Granger causality that provide means to establish directional relations between the brain regions. For instance, it can help verify whether the observed associations were indeed stemmed from causal (i.e., in its purely statistical term) relations among these regions. Furthermore, it can enable researchers to investigate the existing theorems and hypotheses about the brain functioning and its regional interactivity in a more robust way. For example, G-causality can be adapted for the study of the affect in terms of brain’s regional functional connectivity, thereby allowing for reconciliation among the contrasting results of the bipolarity [18], bivalent [19,20,21], and affective workspace [2,22] hypotheses [2,26,27,28,29,30]. To the best of our knowledge, no previous study has considered the use of Granger causality for this purpose.

Our contributions are threefold. First, we show that the Negative, Neutral, and Positive affects emerge from subtle variation in information flow of the brain regions that are in both hemispheres. Second, we show that these regions are common between these affect states than distinct to a specific affect. Third, we show that these regions are characterized with both short- as well as long-range information flow. This provides evidence for the presence of simultaneous integration and differentiation in the brain functioning that leads to the emergence of different affects. Taken together, our findings appear to be more in line with the affective workspace hypothesis [2,22] than the bipolarity [18] or the bivalent hypotheses [19,20,21]. Our results are also in line with the findings on the presence of intrinsic large-scale interacting brain networks that underlie the production of psychological events [69,70,71,72]. We believe that our study can help advance our understanding of the neural basis of the human’s emotions by identifying the signatures of differential affect in subtle variation that occurs in the whole-brain cortical flow of information.

In regards to our study, there is an important point that deserves further clarification. A number of neuroscientific findings argue that the basic emotions are localized to the firing of subcortical circuits [13,73]. They also show that the emergence of the affect rather originates from these subcortical regions [73] where multiple brainstem-derived modulatory neurotransmitters contribute to emotion and emotional behavior [74,75,76]. In this respect, it is crucial to note that the present study is not about the origin of the affect in such subcortical levels. It primarily aims at the higher-level cortical regions to verify whether these regions are common/distinct to/between the Negative, Neutral, and Positive affects. Subsequently, it investigates the extent to which the potentially differential flow of information among these regions can account for neural substrates of the Negative, Neutral, and Positive affects.

## 2. Materials and Methods

### 2.1. The Dataset

SEED [65] corresponds to sixty-two-channel EEG recordings (Figure 1B) of fifteen Chinese subjects (7 males and 8 females; Mean (M) = 23.27, Standard Deviation (SD) = 2.37). All participants were right-handed (with self-reported normal or corrected-to-normal vision and normal hearing) and were students from Shanghai Jiao Tong University. They watched fifteen Chinese movie clips (four minutes in duration) that elicited Negative, Neutral, and Positive affects. These individuals were selected based on the Eysenck Personality Questionnaire (EPQ) [77] personality traits. EPQ evaluates the individuals’ personality along three independent dimensions of temperament: Extraversion/Introversion, Neuroticism/Stability, and Psychoticism/Socialization. Eysenck et al. [77] reported that it appears that not every individual can elicit specific emotions immediately (even in the presence of explicit stimuli). They also reported that the individuals who are extraverted and have stable moods tend to elicit the right emotions throughout the emotion-based experiments. Therefore, the authors in SEED adapted the same personality criteria that was reported by Eysenck et al. [77] to select the fifteen individuals that participated in their experiment.

Prior to a SEED experiment, the authors asked twenty volunteers to assess a pool of movie clips in a five-point scale. Based on the result of this assessment, they selected the fifteen movie clips (i.e., five clips per Negative, Neutral, and Positive affects) whose average score were ≥ 3 and ranked in the top 5 in each affect category. They further verified that the selected movie clips indeed elicited the targeted affect in a follow-up study [79] that included nine separate individuals who were different from the twenty volunteers that originally involved in rating and selection process of fifteen movie clips. The authors then used these movie clips in SEED experiment.

In SEED, each experiment included (Figure 1A) a total of fifteen movie clips per participant. In this setting, each movie clip was proceeded with a five-second hint to prepare the participants for its start. This was then followed by a four-minute movie clip. At the end of each movie clip, the participants were asked to answer three questions that followed the Philippot [78]. These questions were the type of emotion that the participants actually felt while watching the movie clips, whether they previously watched the original movies from which the clips were taken, and whether they understood the content of those clips. The participants responded to these three questions by scoring them in the scale of 1 to 5. The participants were then instructed to take a fifteen-second rest before the next movie clip in the experiment started. Each individual participated in three experiments with an interval of about one week between them. The same set of fifteen movie clips were used in all of these three experiments. Every participant watched the same set of fifteen movie clips in the same order of their presentations.

The movie clips within each experiment were ordered in such a way that two clips with the same emotional content (e.g., both targeting Negative affect) were not presented consecutively to the participants. Additionally, these clips were selected based on the criteria that their lengths were not too long to induce fatigue on the subjects while watching them that their contents were easy to understand by the participants without any explicit explanation, and that each clip elicited a single desired target affect (e.g., Negative or Positive).

SEED comes with its preprocessed EEG recordings which we used in the present study. Its preprocessing steps consist of downsampling the EEG recordings to 200 Hz followed by bandpass filtering the signals within 0–75 Hz. These steps were applied on the extracted EEG segments that corresponded to the duration of each movie clip. Further details on SEED experiment, EEG channels’ arrangement, movie clips’ selection criteria, data acquisition, and preprocessing, labeling the emotional states associated with each movie clip, etc. can be found in reference [65] and the following link (http://bcmi.sjtu.edu.cn/~seed/seed.html).

### 2.2. Data Selection and Validation

In our study, we considered only the first experiment of every participant (i.e., out of their three times participating) and included all of its corresponding fifteen movie clips trails (Figure 1A). We also considered all sixty-two channels of EEG recordings of these individuals. Prior to any further analyses, we validated the selected data through the following steps.

In our study, we considered only one session of every participant (per affect) out of their three times participations [65]. This resulted in a total of 15 participants × 3 affect states. We considered all sixty-two channels of EEG recordings of these individuals. Prior to any further analysis, we validated the selected data through the following steps.

First, we ensured that all EEG recordings that were included in our study were sufficiently long (Mean (M) = 45,286.71, Standard Deviation (SD) = 2776.61, CI${}_{95\%}$ = [44,565.70 46,007.73], minimum = 37,001, maximum Length = 47,601) where CI${}_{95\%}$ refers to the 95% confidence interval. We then trimmed all participants’ EEGs to have equal lengths of 37,000 data points (i.e., the minimum length of EEG recording observed in the participants’ data).

Next, and prior to any further data validation, we performed two necessary preprocessing steps on all participants’ EEG data: detrending (i.e., subtracting the best fitting line) and z-standardization (i.e., subtracting the ensemble mean and dividing by ensemble standard deviation from each time series to obtain data with zero-mean and unit standard deviation) [80]. These steps help remove nonstationarity in the mean and the standard deviation (i.e., nonstationarity that is reflected in variance over time of the time series mean and standard deviation).

We then performed Augmented Dickey Fuller (ADF) [81] and Kwiatkowski–Phillips–Schmidt– Shin (KPSS) [82] tests on all EEG channels (per participant, per affect) to ensure that they were covariance stationary and subsequently marked those participants’ data whose EEG channel(s) did not pass these tests. During these tests, we noticed that all three sessions of one of the participant did not satisfy the requirement for covariance stationarity (i.e., trend–stationarity that implies mean and variance do not change over time). Therefore, we did not include this participant in the further analyses. We also observed that two of the participants’ EEG recordings from their first sessions did not pass these tests on two of their EEG channels. Therefore, we replaced these participants’ first session EEG with their corresponding second and third sessions, respectively, that passed these tests on all of their EEG channels.

### 2.3. Causal Density and Causal Flow Computations

In this study, we adapted time-domain Granger causality, given its known statistical properties that allows for straightforward tests of significance [58,59]. A crucial choice of parameter in G-causality analysis is its model order i.e., the number of previous observations to consider while estimating the autoregressive model [59]. Given the fact that different model orders may lead to different results, we used Akaike and Bayesian information criteria (AIC and BIC, respectively) to determine the model order for the participants’ EEG time series. Whereas BIC showed inconsistencies in the choice of model order for different EEG time series (M = 5.71, SD = 0.83, CI${}_{95.0\%}$ = [5.32 6.11]), we found that AIC considered the model order = 12 for all the participants’ validated time series. Therefore, we used model order = 12 in the subsequent steps.

We utilized the Granger causality toolbox by Barnett and Seth [58] to compute (per participant, per affect) unit (i.e., per EEG channel) causal density and unit causal flow. Causal density (cd) expresses the overall degree of causal interactivity [59]. It is defined as the mean of all pairwise G-causalities between system elements, conditioned on the remainder of the system. The causal density of a system *X* (e.g., all the EEG channels in our case) is computed as (Seth [59], p. 268):(1)$$\begin{array}{ccc} cd(X) & = & \frac{1}{n(n-1)}\sum_{i\neq j}F_{X_{i}\to X_{j}|X_{\left[ij\right]}} \end{array}$$
and $X_{\left[ij\right]}$ denotes the subsystem of *X*, where elements $X_{i}$ and $X_{j}$ are excluded. The unit causal density (ucd) of any $X_{i}\in X$ (e.g., a single EEG channel) is then the summed causal interactions that involves $X_{i}$, normalized by ∥X∥ i.e., number of elements of *X* (ibid). This, in turn, results in *n* ucd values for ∥X∥=n (in our case, 62 ucd values for 62 EEG channels, per participant, per affect).

On the other hand, the unit causal flow (ucf) of an element $X_{i}\in X$ is defined as the difference between its in-degree and out-degree [59]. In other words, ucf of $X_{i}\in X$ (e.g., *i*th EEG channel, 1≤i≤62, in our case) expresses the extent to which $X_{i}$ is influenced by (i.e., in-degree) or influences (i.e., out-degree) the remainder of the elements $X_{j}\in X,\;j\neq i,1\leq i,\;j\leq 62$ (e.g., all the other EEG channels in our case).

The causal density [58] is the measure of a system (i.e., whole-brain in our case) dynamical complexity. A high causal density reflects simultaneous integration and differentiation in network dynamics. Precisely, it indicates that the elements within a system (i.e., each EEG channel in our case) are both globally coordinated in their activity (in order to be useful for predicting each other’s activity) while being dynamically distinct (so that different elements contribute in different ways to these predictions) [83,84]. In this respect, the unit causal density then refers to contribution of each of the system’s element (i.e., each EEG channel in our case) to this overall causal density. In the same vein, the unit causal flow of a system’s element (i.e., each EEG channel in our case) identifies its distinctive causal effects: an element with highly positive causal flow exerts a strong causal influence on the system as a whole [59].

While computing the EEG channels’ unit causal density and causal flow, per participant, per affect, we applied the Durbin–Watson test [85] to ensure that the model’s residuals were uncorrelated. We verified this by observing that all adjusted r-squared were above their empirical limit of > 0.3 [59,85] (M = 0.90, SD = 0.06, CI = [0.88 0.91]). We also applied the consistency test [80] and observed that it was above its empirically accepted limit of > 0.80 [59,80] (M = 85.29, SD = 6.89, CI = [80.91 89.68]).

### 2.4. Statistical Analysis

The EEG time series that were considered in this study had a relatively large sampling rate (i.e., 200 Hz after downsampling) and corresponded to a four-minute experimental setting. As a result, they were long enough to allow for the choice of such dynamical analysis as Granger causality, given the sensitivity of such techniques to the length of time series. In addition, EEG has a higher temporal resolution (i.e., in comparison with other neuroimaging techniques such as fMRI and fNIRS) and therefore poses itself as a better choice for the study of causal connectivity (i.e., in a pure statistical than anatomical sense). Detailed discussion on such issues and considerations can be found in [50,51,58,59,62,63].

To establish the level of significance for unit causal densities, we first combined the unit causal densities of all EEG channels of all participants in Negative, Neutral, and Positive affects states and performed a one-sample test of significance (10,000 simulation runs) at a 95.0% confidence interval on them (M = 0.69, SD = 0.30, CI${}_{95.0\%}$ = [0.68 0.70]). Subsequently, we only considered the values that were >0.70 i.e., unit causal densities that were above the upper bound of the bootstrap test of significance at a 95.0% confidence interval.

We carried out three different analyses. They were (1) Spearman correlation to verify whether there was a correspondence between the unit causal densities associated with Negative, Neutral, and Positive affects; (2) test of significance to determine whether unit causal densities and unit causal flows (i.e., all EEG channels) differed significantly between these affect states; and (3) determination of the importance of each EEG channel’s unit causal density for differentiating between Negative, Neutral, and Positive affects. In what follows, we elaborate on these steps.

#### 2.4.1. Correlation

We computed the Spearman correlations between the whole-brain unit causal density (ucd hereafter) values of every pair of affect states (i.e., Positive versus Neutral, Positive versus Negative, and Negative versus Neutral). We followed this by computing their 95.0% bootstrap (10,000 simulation runs) confidence intervals. For the bootstrap test, we considered the null hypothesis

**Hypothesis** **1 (H1).**
*there was no correlation between every pair of affect states’ whole-brain ucd values.*


Subsequently, we tested Hypothesis 1 against the following alternative Hypothesis.

**Hypothesis** **2 (H2).**
*The whole-brain ucd values of every pair of affect states correlated significantly.*


We reported the mean, standard deviation, and the 95.0% confidence interval for these tests. We also computed the *p*-value of these tests as the fraction of the distribution that was more extreme than the actually observed correlation values. For this purpose, we performed a two-tailed test in which we used the absolute values so that both the positive and the negative correlations were accounted for.

We reported the results of Spearman correlations on unit causal densities that were corrected based on one-sample bootstrap test of significance (10,000 simulation runs) at 95.0% confidence interval in the main manuscript. We provided these results prior to the application of the bootstrap test in Appendix A.

#### 2.4.2. Test of Significance

First, we applied the Kruskal–Wallis test on individuals’ whole-brain (i.e., all EEG channels) ucd values, per affect, which was followed by post-hoc Wilcoxon rank sum tests between every pair of affect (i.e., Positive versus Neutral, Positive versus Negative, and Negative versus Neutral). We further verified these results through application of paired two-sample bootstrap test of significance (10,000 simulation runs) at 95.0% (i.e., p< 0.05) confidence interval. For the bootstrap test, we considered the null hypothesis

**Hypothesis** **3 (H3).**
*The difference between individuals’ whole-brain ucd values in two different affects was non-significant.*


We then tested Hypothesis 3 against the following alternative hypothesis.

**Hypothesis** **4 (H4).**
*The individuals’ whole-brain ucd values significantly differed in two different affect states.*


We reported the mean, standard deviation, and 95.0% confidence interval for these tests.

Next, we performed a channel-wise (e.g., F6 in each affect state) Wilcoxon rank sum test on a unit causal flow of each pair of affect (i.e., Positive versus Neutral, Positive versus Negative, and Negative versus Neutral) to determine whether the information flow from each of EEG channels to the remainder of the channels differed significantly among these affect states. We further verified these results through application of paired two-sample bootstrap test of significance (10,000 simulation runs) at 95.0% (i.e., *p* < 0.05) confidence interval. For the bootstrap test, we considered the null hypothesis**Hypothesis** **5 (H5).**The difference in causal flow from each of EEG channels to the remainder of the channels in three affect states was non-significant.

We tested Hypothesis 5 against the following alternative hypothesis.**Hypothesis** **6 (H6).**The flow of information from EEG channels to the remainder of the channels differed significantly among Negative, Neutral, and Positive affects.

We reported the mean, standard deviation, and 95.0% confidence interval for these tests.

#### 2.4.3. Importance of Channels’ Unit Causal Densities

We used an AdaBoost [86] meta-estimator to determine the utility of EEG channels’ ucd values for quantification of the Negative, Neutral, and Positive affects. This algorithm learns the underlying association among instances of different classes (i.e., Negative versus Neutral versus Positive affect states) in two steps. In the first step, it fits a classifier to the input (a decision tree with depth 1 in our case). In the second step, it fits additional copies of the original classifier to the same input that focuses on misclassified cases, thereby improving its accuracy through readjusting the weights of these more difficult cases. The Adaboost algorithm generates a “feature importance” numeric vector whose values (within [0,⋯,1] interval) specify the critical role of each of the elements of the input data vector (i.e., ucd values in our case) for identification of its corresponding class (e.g., Negative affect). Therefore, we used this algorithm’s “feature importance” vector to determine the utility of EEG channels’ ucd values for quantification of the Negative, Neutral, and Positive affects.

We used individuals’ whole-brain ucd values and formed input feature vectors that were of length sixty-two (i.e., one ucd per channel), per participant, per affect. We then adapted the 1-holdout setting in which we considered data associated with the Negative, Neutral, and Positive affects of a single participant as a test set and used the remaining participants’ data for training the Adaboost meta-estimator. We then tested the model’s performance on the holdout data. We repeated this procedure for every participant. This resulted in fourteen different test cases in which we used to compute the model’s accuracy, precision, recall, and confusion matrix.

Next, we applied a one-sample bootstrap test of significance (10,000 simulation runs) at 95.0% confidence interval on the Adaboost’s calculated feature importance. We then trimmed the individuals’ ucd vectors (i.e., input data to Adaboost) to only include ucd values whose calculated feature importance by Adaboost were within or above feature importance’s 95.0% confidence interval. We followed the same training and testing strategy as in the case of whole-brain ucd values and reported the model’s accuracy, precision, recall, and confusion matrix. Given the three-affect setting in which every participant had an equal number of Negative, Neutral, and Positive affects data, the chance-level accuracy was ≈33.33%.

Finally, we used the Adaboost’s accuracies in the case of whole-brain ucd values versus ucd values whose importance were within or above feature importance’s 95.0% confidence interval and applied the Wilcoxon rank sum test on them to determine whether utilization of the whole-brain causal information bore a significant difference in quantification of Negative, Neutral, and Positive affects states. We further verified these results through application of paired two-sample bootstrap test of significance (10,000 simulation runs) at 95.0% (i.e., *p* < 0.05) confidence interval. For the bootstrap test, we considered the null hypothesis**Hypothesis** **7 (H7).**The significance of the use of whole-brain ucd values for classification of different affect states was non-significant.

We tested Hypothesis 7 against the following alternative hypothesis.**Hypothesis** **8 (H8).**The use of whole-brain ucd values had a significant effect on classification of different affect states.

We reported the mean, standard deviation, and 95.0% confidence interval for these tests.

For the Kruskal–Wallis test, we reported the effect size $r=\sqrt{\frac{\chi^{2}}{N}}$ [87] with *N* denoting the sample size and $\chi^{2}$ is the respective test-statistics. In the case of Wilcoxon tests, we used $r=\frac{W}{\sqrt{N}}$ [88] as effect size with *W* denoting the Wilcoxon statistics and *N* is the sample size. All results reported were Bonferroni corrected. All analyses were carried out in Matlab 2016a (The MathWorks, Inc., Natick, Massachusetts, USA) and Python 2.7 (Python Software Foundation. Python Language Reference, version 2.7., available at the following link (http://www.python.org). For classification of the affect states and determination of ucd values’ importance, we used Python’s scikit-learn [89] multi-class implementation of Adaboost algorithm (referred to as AdaBoost-SAMME [90]) that utilizes a decision tree with depth 1 as its base classifier.

With regard to our analyses, there are two points that are worth further clarification. They are the choice of non-parametric tests and the follow-up bootstrap test of significance. Prior to our analyses, we checked the participants’ unit causal density and the unit causal flow in each of the Negative, Neutral, and Positive affects states (separately as well as combined, with respect to the both individuals and the EEG channels for each of the affect). We found that they did not follow normal distribution. Therefore, we opted for non-parametric analyses. In the case of bootstrap, on the other hand, we realized that our analyses were performed based on a small sample of participants (i.e., fourteen individuals). We also observed that our analyses of the participants’ unit causal flow yielded small effect sizes. Therefore, it was crucial to ensure that any significant results that we observed in our analyses were not due to a subsample of individuals (i.e., distorted data and hence lack of central tendency). This concern was further strengthened by the result of the non-normality of the participants’ unit causal density and the unit causal flow. Therefore, we decided to also apply the bootstrap test (i.e., random sampling with replacement) that was carried out for 10,000 simulation runs (therefore allowing for potential outliers to be repeated with higher probability and more frequently) at 95% confidence interval (i.e., *p* < 0.05 significance level), thereby enabling us to further verify our results.

### 2.5. Ethics Statement

This study was carried out in accordance with the recommendations of the ethical committee of the Advanced Telecommunications Research Institute International (ATR). The protocol was approved by the ATR ethical committee (approval code: 17-601-4).

## 3. Results

### 3.1. Correlation

We found a significant correlation between Positive and Negative (Figure 2A, left column, *r* = 0.58, *p* < 0.00001), Positive and Neutral (Figure 2B, left column, *r* = 0.58, *p* < 0.00001), and Negative and Neutral (Figure 2C, left column, *r* = 0.53, *p* < 0.00001). These correlations that were also verified by the results of their corresponding bootstrap test of significance (10,000 simulation runs) at 95.0% confidence interval (Figure 2A–C, right column) were stronger between Positive and Negative as well as Positive and Neutral than Negative and Neutral. Table 1 summarizes the results of these bootstrap test of significance (for the results prior to the application of the bootstrap test to determine the ucd values’ significant level, see Appendix A).

### 3.2. Unit Causal Density

Figure 3A illustrates the grand averages of the spatial map of ucd values in Negative, Neutral, and Positive states. These subplots indicate an incremental pattern of ucd values from the Negative to Positive affect that is distributed over the whole-brain EEG recordings. Arrangement of the EEG channels associated with these ucd values is shown in Figure 3B.

A Kruskal–Wallis test indicated a significant difference between the whole-brain unit causal densities associated with Negative, Neutral, and Positive affects states (*p* < 0.001, H(2, 2603) = 16.15, *r* = 0.09). A post-hoc Wilcoxon test (Figure 3C) further identified that, whereas a Positive affect was associated with a higher whole-brain unit causal densities than the Negative (*p* < 0.001, W(1734) = 3.84, *r* = 0.10, M${}_{Positive}$ = 0.49, SD${}_{Positive}$ = 0.49, M${}_{Negative}$ = 0.40, SD${}_{Negative}$ = 0.50) and Neutral (*p* < 0.01, W(1734) = 2.86, *r* = 0.07, M${}_{Neutral}$ = 0.42, SD${}_{Neutral}$ = 0.49), such a difference was non-significant between the Negative and Neutral affect states (*p* = 0.31, W(1734) = 1.00, *r* = 0.02).

Figure 4 shows the results of the paired two-sample bootstrap test (10,000 simulation runs) at 95.0% confidence interval (CI). This figure confirms that the participants’ whole-brain ucd values in the Positive state was significantly higher than their Neutral (Figure 4A) and Negative (Figure 4B) states. It also indicates that the difference between the whole-brain ucd values in Negative and Neutral states was non-significant (Figure 4C). Table 2 summarizes these results.

### 3.3. Unit Causal Flow

Figure 5, Figure 6 and Figure 7 visualize the unit causal flow in Negative, Neutral, and Positive affects states whose values are scaled within [0,⋯,1] for better comparison. In these figures, each subplot depicts the flow of information from a given EEG channel (e.g., F3) to the remainder of channels. These figures verify that the flow of information was associated with both hemispheres. They also identify a number of channels that exhibited short- as well as long-range information flow. This indicated the presence of cross-hemispheric whole-brain information flow and communication that was independent of the affect.

In the case of Negative affect (Figure 5), F5’s and FC5’s (i.e., left-hemispheric channels) information flow extended to the right parietal and occipital regions. Similarly, FT7 appeared to influence the entire left hemisphere. Other examples in the case of Negative affect included the information flow from FC6 to the right-hemisphere central, parietal, and occipital regions, F8’s information flow to the left frontal region, F6’s information flow to the left parietal and occipital regions, CZ’s bi-hemispheric frontal flow of information, CPZ’s bi-hemispheric parietal and occipital influence, and the CB2’s information flow that extended bi-hemispherically to the frontal, central, and parietal regions.

Similarly, in the Neutral affect (Figure 6), we observed the extent of F6’s information flow to the parietal and occipital regions. We also observed that F8’s information flow extended bi-hemispherically to frontotemporal and FC5’s flow of information to the right hemisphere’s temporal and occipital. The information flow from FC6 appeared to extend to the right-hemisphere central, parietal, and occipital regions. Moreover, CZ exhibited a bi-hemispheric frontal information flow and CP5’s information flow was extended to the right-hemispheric frontal, central, and parietal. T8’s information flow appeared to reach to the left-hemispheric temporal region. Finally, CB2’s information flow extended bi-hemispherically to the frontal, central, and parietal regions.

Considering the Positive affect (i.e., Figure 7), F8, FT7, FC5, FC6, FC8, C5, CZ, C6, CP5, and CB2 are the channels that exhibited such bi-hemispheric, short- and long-range information flow. Specifically, we observed the F8’s influence on a left frontal area, FT7’s information flow to the entire left-hemisphere, and FC6’s information flow to the left parietal and occipital regions. Additionally, CZ’s information showed influence on bi-hemispheric frontal, parietal, and occipital areas, and CB2 exhibited a bi-hemispheric information flow to the frontal, central, and parietal regions.

In the case of Positive versus Neutral, a channel-wise paired Wilcoxon rank sum test (Appendix B, Table A2) identified that F5, F4, F6, FC3, C2, C4, CPZ, CP6, and TP8 had significantly higher information flow in the Neutral than Positive states (i.e., Appendix B, Table A2, entries Neutral > Positive). On the other hand, FT7, FT8, TP7, CP3, P5, P1, P4, P6, P8, PO5, POZ, PO6, OZ, and O2 were the channels in the case of Positive affect that had higher flow of information (i.e., Appendix B, Table A2, Positive > Neutral entries). For Positive versus Negative affect states (i.e., Appendix B, Table A3), information flow was higher in the case of Negative than Positive (i.e., Negative > Positive entries) in channels F5, F3, F6, FT7, FC3, FC2, C3, CP3, CPZ, CP6, and O2. On the other hand, Positive affect was associated with higher information flow than the Negative affect (i.e., Appendix B, Table A3, Positive > Negative entries) in channels FP1, FPZ, F7, F1, F2, F4, C4, T8, TP7, CP2, CP4, P7, P5, P3, P1, P2, P4, PO5, PO8, and OZ. With respect to Negative versus Neutral (Appendix B, Table A4), we observed that Neutral affect showed significantly higher information flow than Negative affect in channels FPZ, F1, F2, F4, FCZ, C4, T8, and CP2 (Appendix B, Table A4, entries Neutral > Negative). Similarly, the Negative affect was characterized with significantly higher information flow in (Appendix B, Table A4, entries Negative > Neutral) F3, FZ, FT7, FT8, CP3, P5, P4, P6, PO5, PO3, POZ, PO4, PO6, CB1, and O2. Further details on channel-wise comparison of the information flow in Negative, Neutral, and Positive affects are presented in Appendix C.

### 3.4. Importance of Channels’ Unit Causal Densities

Figure 8A shows the spatial map of the Adaboost’s feature importance associated with sixty-two EEG channels’ ucd values. This subplot identifies that the ucd values associated with higher feature importance to distinguish between Negative, Neutral, and Positive affects were distributed in multiple brain regions that included the frontal, central, parietal, and occipital area. The EEG channels’ arrangement associated with these feature importance values are shown in Figure 8A, right subplot.

Figure 8B shows the affect-wise confusion matrix of the Adaboost classifier for predicting the Negative, Neutral, and Positive affects states using all EEG channels’ ucd values. The one-sample bootstrap test of significance (10,000 simulation runs) at 95.0% confidence interval indicated that the overall model performance on individuals’ data (i.e., 66.67%) was significantly above the chance level (M${}_{accuracy}$ = 57.17, SD${}_{accuracy}$ = 4.04, CI${}_{accuracy}$ = [50.0 64.29], chance level ≈ 33.33%). We also observed that the model’s predictions were substantially higher for Negative (accuracy = 78.57%, precision = 0.79, recall = 0.58) and Positive (accuracy = 71.43%, precision = 0.71, recall = 0.91) than the Neutral affect (accuracy = 50.00%, precision = 0.50, recall = 0.58). In addition, Figure 8B indicates that the Neutral affect was mostly misclassified as a Negative rather than a Positive affect.

We next considered the EEG channels whose ucd values’ corresponding feature importance were within or above 95.0% confidence interval of the result of the one-sample bootstrap test of significance (10,000 simulation runs) on these feature importances (M = 0.13, SD = 0.03, CI${}_{95.0\%}$ = [0.08 0.21]). This allowed us to determine whether the use of whole-brain information flow bore a substantial effect on the accuracy of the classifier. Figure 8C shows the affect-wise confusion matrix of the Adaboost classifier for predicting the Negative, Neutral, and Positive affects states when only the EEG channels’ ucd values with their feature importance ≥ 0.08 were considered for model training. A comparison between Figure 8B,C verifies the substantial reduction of the model’s accuracy when all EEG channels’ unit causal densities were not included (Wilcoxon rank sum: *p* < 0.03, W(26) = 2.43, *r* = 0.46). We also verified this result through the use of two-sample bootstrap test of significance (10,000 simulation runs) at 95.0% confidence interval (M${}_{difference}$ = 22.78, SD${}_{difference}$ = 8.21, CI${}_{difference}$ = [6.10 39.43]).

## 4. Discussion

In this article, we considered the possibility of the emergence of the affect from variation in the whole-brain cortical flow of information. Our study was motivated by the recent results from affective neuroscience that (despite their compelling findings) pointed at contrasting viewpoints on the neural substrates of the affect. Whereas some considered distinct and independent brain systems for the Positive and Negative affects [10,11,18,19,20,21], others proposed the presence of flexible brain regions [15,16,17,22]. We further attributed such a discrepancy in their findings to two primary reasons: (1) their focus on the change in brain activation to identify a specific [31,32,33,34] or subset [27] of the brain regions as sources of different affects. We claimed that this approach could be limited since it neglects the findings that indicate the brain activation and its information content do not necessarily modulate [35]. We also argued that the sole focus on the change in brain activation to realize the neural substrates of the affect is insufficient since the stimuli with equivalent sensory and behavioral processing demands may not necessarily result in differential brain activation [36], (2) their lack of consideration for crucial role of functional interactivity between the brain regions [37]. As a result, they did not take into account the findings that identify the signals from individual cortical neurons are shared across multiple areas and thus concurrently contribute to multiple functional pathways [38].

Subsequently, we utilized the Granger causality [48,49,50,51,52] to analyze the human subjects’ sixty-two-channel EEG recordings [65] who watched movie clips that elicited Negative, Neutral, and Positive affects. We justified our interpretation of Granger causal analysis of the whole-brain in terms of information flow between its regions by observing that Granger causality is an approximation to transfer entropy [55] which itself is nothing but the directional mutual information [53]. An advantage of using Granger causality over transfer entropy is that, unlike the latter’s complicated estimation [55,56,57], the known statistical properties of the Granger causality allows for straightforward tests of significance [58,59]. In this regard, an important consideration that deserves restating [91] is that the Granger causality is a statistical formulation of causality and, as such, a significant interaction measured by this model does not by itself imply the presence of a corresponding physical interaction.

We found that the different affect states were associated with the flow of information that was present in the both hemispheres. These results were in line with the findings that identified the brain bilateral activation during story comprehension [92,93] as well as watching movies [94,95]. They also further complemented these findings by identifying the presence of a corresponding bilateral flow of information associated with the change in the brain activation in response to Negative, Neutral, and Positive affects states. Previous research [96] also identified the activation of a distributed network of the brain areas in response to basic emotions, thereby suggesting that the distributed emotion-specific activation patterns may provide maps of internal states that correspond to specific subjectively experienced, discrete emotions [97]. Our results extended these findings by presenting evidence that identified the varying degree of information flow among the elements of such a distributed brain network may explain the emergence of different affects.

Our findings appeared to be more in line with the affective workspace hypothesis [2,22] than the bipolarity [18] or the bivalent hypotheses [19,20,21]. For instance, the bipolarity hypothesis [18] attributes the Positive and Negative affects to the opposing ends of a single dimension [23,24]. As a result, it is plausible to expect anti-correlations between these affect states’ flow of information. Contrary to this expectation, we observed positive correlations among them. On the other hand, the bivalent hypothesis [19,20,21] emphasizes the presence of two distinct and independent brain systems for the Positive and Negative affects [19,20,21]. This makes it plausible to expect that the information flow associated with these affect states to form non-overlapping and distinct patterns. Although our analyses identified a number of brain regions whose unit causal flows significantly differed between Negative, Neutral, and Positive affects (see Appendix C), we also observed that these regions were common between these affects. Additionally, the observed flow of information among distributed brain regions that were common between the Negative, Neutral, and Positive affects was also in accordance with the affective workspace hypothesis that expresses that the differential affects are the brain states that are supported by flexible than consistently specific set of brain regions [25]. From a broader perspective, our results resonated with the findings that emphasize the importance of the functional connectivity between distributed brain regions that include pre/frontal, parietal, premotor and sensory, and occipitotemporal regions [98] and the implication of such large-scale and distributed networks in the brain functions [99,100,101].

Further evidence in support of this view came from the classification results of these affect states in which a simple linear model was sufficient to significantly distinguish between them. These results were in line with our correlation analyses in that they implied that any change in the information flow in one affect can be explained in terms of a linear change in the corresponding flow of information in the other affect state that was in the same direction as of the first one. We also observed that a linear model that only utilized the regions with a significantly different flow of information, yielded a significantly poorer performance in comparison with the setting in which the whole-brain cortical information flow was considered. Additionally, the mere use of these regions appeared to cause the linear model to treat all three affects as Neutral. This interpretation can be verified by observing the sudden and substantial increase in linear model’s correct classification of the Neutral affect that was accompanied by its significantly reduced accuracy in the case of both Positive and Negative affect states. In fact, the recurrence of some of these regions in such domains as social cognition and theory of mind [102], story comprehension [92,93], autobiographical memory [103], decision-making [104], working memory [105], and self-referential processing [106], along with the reduced accuracy of the linear model in our study further implied that [1] these regions may not be specific to the affect but may constitute to other cognitive and perceptual events, thereby forming domain-general networks [107,108]. Considering these observations, our findings indicated that the whole-brain flow of information that was manifested in distributed cortical regions were not only shared among the Negative, Neutral, and Positive affects states but also was able to best distinguish between them.

The dynamical system analysis [39,40] of the brain ongoing activity [43,47] considers its neural dynamics and the functional connectivity that accompanies these neural activities as two aspects of the brain’s information processing. It holds that the more complex neural activity is characterized with an increased functional connectivity [109]. In this view, such an increase may represent the short- and long-range information processing across the brain regions [45,109,110,111,112]. These factors that underline [113] the presence of “differentiation” (i.e., the presence of subsets of a system that are dynamically distinct) and “integration” (i.e., the presence of coherence in such a system as a whole) appear to constitute the cognitive and behavioral flexibility of a system to respond specifically and selectively to a broad range of stimuli [114,115]. Our results extended these findings to the case of neural substrates of different affects. Specifically, they identified the emergence of different affects from subtle variation in the flow of information between various brain regions that exhibited both short-range (i.e., local) interaction with their neighboring brain areas and long-range (i.e., distributed) communication with the brain regions that were in both hemispheres. This is due to the observation that the causal density [58] reflects simultaneous integration and differentiation in which elements within a system are both globally coordinated in their activity (in order to be useful for predicting each other’s activity) while being dynamically distinct (so that different elements contribute in different ways to these predictions) [83,84]. Therefore, the unit causal density of each of these elements quantifies their respective contribution to the information flow in the system as a whole [59]. Our results also provided further supporting evidence for the thesis [18,116,117] that point at the emerging nature of different affects. This viewpoint explicitly assumes that emotions cannot be merely redefined as their ingredients [118]. It subsequently relates their emergence to causal association among their underlying neural activity [11].

Although we noticed that a subset of channels contributed more to such distributed patterns, the classification results of the Negative, Neutral, and Positive affects indicated that it was the whole-brain information flow that yielded a significantly higher prediction power for distinguishing between these states. Our results that complemented the fMRI findings based on a brain activation pattern [96] revealed that the variational pattern of EEG-based flow of information among anatomically distributed brain regions contained the most accurate neural signature of individuals’ mental states that underlined their discrete affect. They were also in accordance with Farroni et al. [7] and Watson and Tellegen [3] that showed that the joint activity from multiple regions discriminated best between different emotions. In this respect, our results hinted at observations that the large-scale cortical networks are crucially involved in representing such high-level mental states that form the foundations for describing the distinctively elicited emotions [2,4]. They also indicated that features from these cortical regions may contribute differentially during the concepts’ categorization [119].

In this article, we provided evidence for the possibility of the emergence of the affect from the variation in the information flow among distributed brain regions that were located in both hemispheres. We showed that these regions were not distinct to a specific affect and that they were characterized with both short- as well as long-range information flow and communication. This provided evidence for the presence of simultaneous integration and differentiation in the brain functioning that leads to the emergence of different affects. These results were in line with the findings on the presence of intrinsic large-scale interacting brain networks that underlie the production of psychological events [69,70,71,72]. They can help advance our understanding of the neural basis of the human’s emotions by identifying the signatures of differential affect in subtle variation that occurs in the whole-brain cortical flow of information.

## 5. Conclusions

In this article, we provided evidence for the possibility of the emergence of the affect from the variation in the information flow among distributed brain regions that were located in both hemispheres. We supported this viewpoint by showing three results: (1) the whole-brain cortical flow of information was positively correlated between these affect states; (2) although these distributed regions were shared among the Negative, Neutral, and Positive affects, they appeared to share information differentially in response to these affect; (3) a simple linear model was able to distinguish between these affect states with a significantly above average accuracy when it used the whole-brain cortical flow of information. For instance, whereas we observed a non-significant difference between the channels’ causal densities in the case of Negative and Neutral affect, we found a considerable number of channels’ whose causal flows differed significantly in response to these two affect states. These results hinted at the possibility of interpreting the cortical responses to these affect in terms of the variation in the flow of information that differed significantly among these channels. In other words, the observed differences suggested that although these channels were common between the Negative and the Neutral affect and that their causal densities (i.e., the overall degree of causal interactivity) were non-significant, their causal flows (i.e., the extent of their influence on/by the other channels) were significantly varying in response to these differential affect.

On the other hand, a puzzling observation with regards to the linear classifier was the higher percentage of misclassification between the Neutral and the Negative affect. We noticed that this could not readily be attributed to the sample size since we used a balanced dataset in our analyses (i.e., equal number of samples for each of the Negative, Neutral, and Positive affects, per participant). Another possible reason behind the observed effect could have been the difference in the shared information between these affect. However, the results of the correlation analyses (i.e., linear measure of mutual information [120]) did not provide any further insight on this matter. For instance, these results could not explain the higher misclassification of the Neutral as the Negative than the Positive affect, despite the fact that the Neutral affect appeared to share slightly higher information (i.e., comparably higher correlation) with the Positive than the Negative affect. Furthermore, the observed correlations between Positive and the other two affect states were equivalent. As a result, if the higher shared information was to blame, then it should have caused a comparable misclassification between the Positive and the Negative, which evidently was not the case. Conversely, the lower shared information also was not sufficient to account for the observed higher misclassification of the Neutral affect as the Negative than the Positive affect. This is due to the observation that the high rate of misclassification between the Neutral and Negative should have in principle been the lowest, given their lowest correlation among the three pairwise comparisons.

The implications of these observations were threefold. First, although the Negative, Neutral, and Positive affects appeared to share a common distributed neural correlates, the underlying dynamics of such a whole-brain cortical information flow might be shared differentially among them. Second, such a dynamics might have more in common between the Negative and the Neutral affect states than the Neutral and the Positive or the Negative and the Positive affect. Third, this potential dynamics that governed the cortical responses to these affect might not primarily be explained through a linear modeling of such a common and distributed neural activity. Therefore, future research that takes into account the nonlinear dynamics of neural activity [55,121] in response to Negative, Neutral, and Positive affects is necessary to further extend our understanding of the potential causes of the observed dis/similarities between these affect states.

Many neuroscientific studies are based on a small number of participants [122,123,124,125]. For instance, the recent comprehensive meta-analysis by Lindquist [2] that reported on an extensive coverage of 914 experimental contrasts in affect-related studies accounted for 6827 participants (i.e., 6827/914 ≈ 7.47 participants on average). Furthermore, most of the individuals that are included in these studies share the same geographical and/or cultural background (but also see [95] for a small deviation). Neuropsychological findings indicate that the individuals’ ability to experience pleasant/unpleasant feelings to express these subjective mental states in terms of such attributes as positive or negative to be the unifying and common concept across cultures [8,9]. However, future research that includes a larger human sample as well as different age groups and more cultural diversity is necessary for drawing a more informed conclusion on the findings that were presented in this article.

Finally, it would also be interesting to extend the results in the present study to the case in which the cortical activities are simultaneously acquired alongside the brain subcortical responses to the affect. [13,73]. Such studies can shed light on the underlying mechanisms that lead to the observed distributed variation in the whole-brain cortical information flow, thereby allowing for the better understanding of the correspondence between these cortical information sharing and their underpinning neuromodulatory mechanisms and dynamics [74,75,76].

## Figures and Tables

**Figure 1 brainsci-10-00008-f001:**
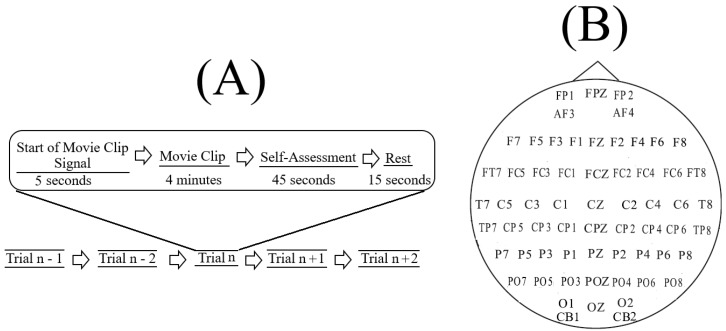
(**A**) schematic diagram of an experiment as described in [65]. Each experiment included a total of fifteen movie clips (i.e., *n* = 15, audiovisual), per participant. In this setting, each movie clip was proceeded with a five-second hint to prepare the participants for its start. This was then followed by a four-minute movie clip. At the end of each movie clip, the participants were asked to answer three questions that followed the Philippot [78]. These questions were the type of emotion that the participants actually felt while watching the movie clips, whether they watched the original movies from which the clips were taken, and whether they understood the content of those clips. The participants responded to these three questions by scoring them in the scale of 1 to 5; (**B**) arrangement of the EEG electrodes in this experiment. The sixty-two EEG channels were: FP1, FPZ, FP2, AF3, AF4, F7, F5, F3, F1, FZ, F2, F4, F6, F8, FT7, FC5, FC3, FC1, FCZ, FC2, FC4, FC6, FT8, T7, C5, C3, C1, CZ, C2, C4, C6, T8, TP7, CP5, CP3, CP1, CPZ, CP2, CP4, CP6, TP8, P7, P5, P3, P1, PZ, P2, P4, P6, P8, PO7, PO5, PO3, POZ, PO4, PO6, PO8, CB1, O1, OZ, O2, CB2.

**Figure 2 brainsci-10-00008-f002:**
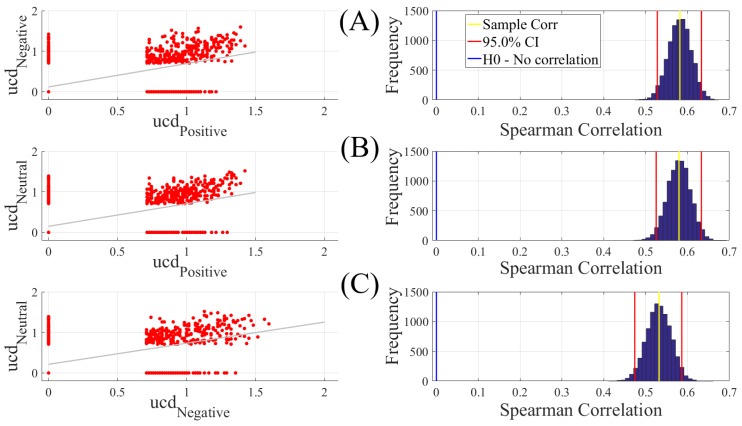
Paired Spearman correlation between participants’ unit causal density (ucd) values (**A**) Positive versus Negative; (**B**) Positive versus Neutral; (**C**) Negative versus Neutral. The subplots on the right column correspond to the bootstrap correlation test (10,000 simulation runs) at 95.0% confidence interval. In these subplots, the zeros correspond to the ucd values that were below the significant level of 0.7 i.e., the upper bound of the one-sample test of significance (10,000 simulation runs) at 95.0% confidence interval (Mean (M) = 0.69, Standard Deviation (SD) = 0.30, Confidence Interval (CI)${}_{95.0\%}$ = [0.68 0.70]). For results prior to the application of the bootstrap test to determine the ucd values’ significant level, see Appendix A.

**Figure 3 brainsci-10-00008-f003:**
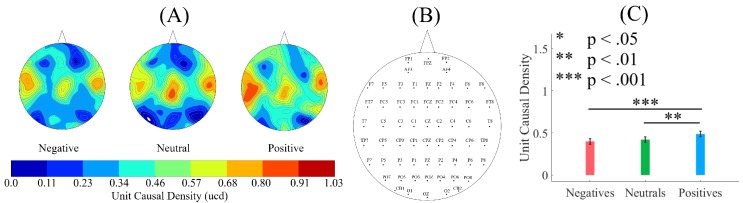
(**A**) grand averages of the spatial map of unit causal density (ucd) in Negative, Neutral, and Positive affects states. Incremental pattern of ucd values from Negative to Positive affect is evident in these subplots; (**B**) EEG channels’ arrangement associated with distribution of ucd values; (**C**) descriptive statistics of the ucd values in Negative, Neutral, and Positive affects states. Asterisks mark the significant differences between these values.

**Figure 4 brainsci-10-00008-f004:**
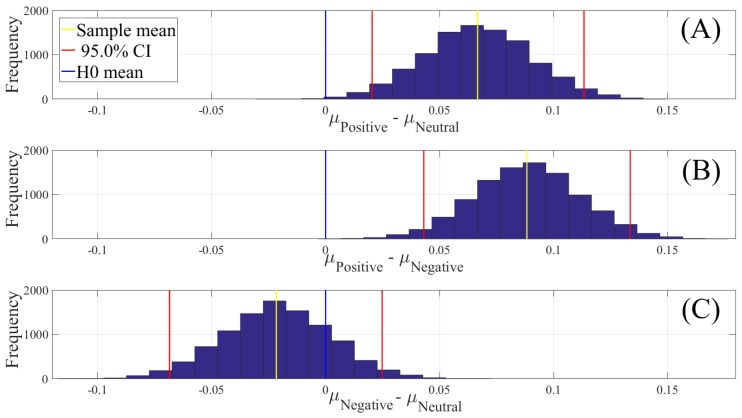
Paired two-sample bootstrap test of significance (10,000 simulation runs) at 95.0% (i.e., *p* < 0.05) confidence interval (CI) associated with the participants’ whole-brain unit causal densities (ucd). Compared pairs of affect are (**A**) Positive versus Neutral; (**B**) Positive versus Negative; and (**C**) Negative versus Neutral. In these subplots, the *x*-axis shows $\mu_{i}-\mu_{j},\;i\neq j$ where *i* and *j* refer to one of the Negative, Neutral, or Positive affect states. The blue line marks the null hypothesis H0 i.e., non-significant difference between the two states’ ucd values. The red lines are the boundaries of the 95.0% confidence interval. The yellow line shows the location of the average $\mu_{i}-\mu_{j},\;i\neq j$ for 10,000 simulation runs.

**Figure 5 brainsci-10-00008-f005:**
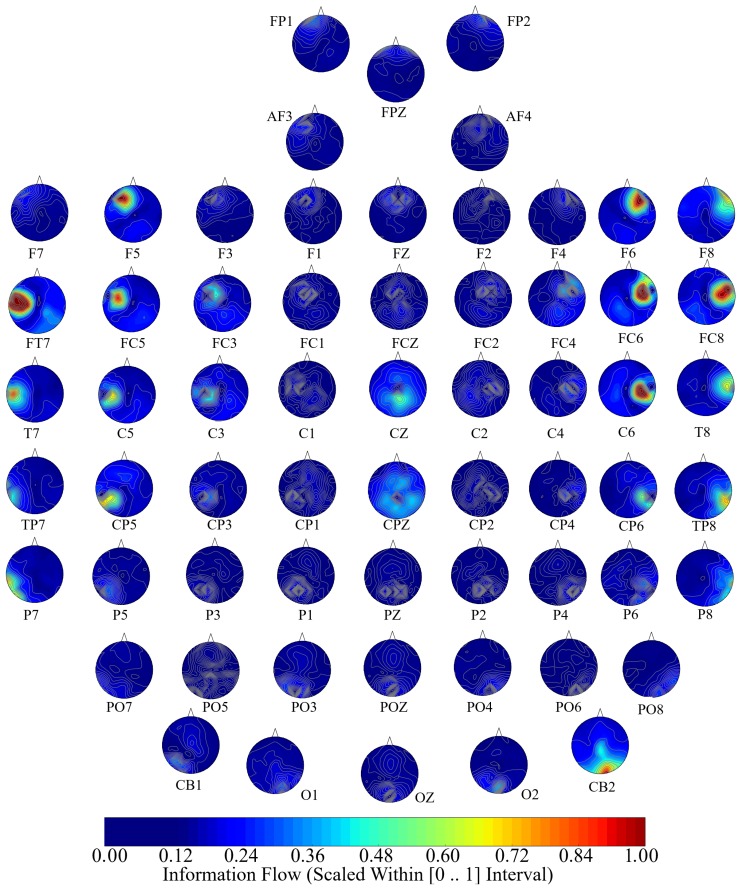
Negative affect’s channel-wise information flow. These subplots identify a bi-hemispheric brain activity in response to Negative affect. They also show that a number of channels are associated with higher short- as well as long-range information (e.g., F5, FC5, FT7, FC6, F8, F6, CZ’, CPZ, CB2). Although these channels appear to have higher local influence in the form of information flow, their corresponding flow of information extend beyond their designated hemispheres, thereby indicating the presence of cross-hemispheric whole-brain information flow and communication. The values in these subplots are scaled within [0,⋯,1] for better comparison.

**Figure 6 brainsci-10-00008-f006:**
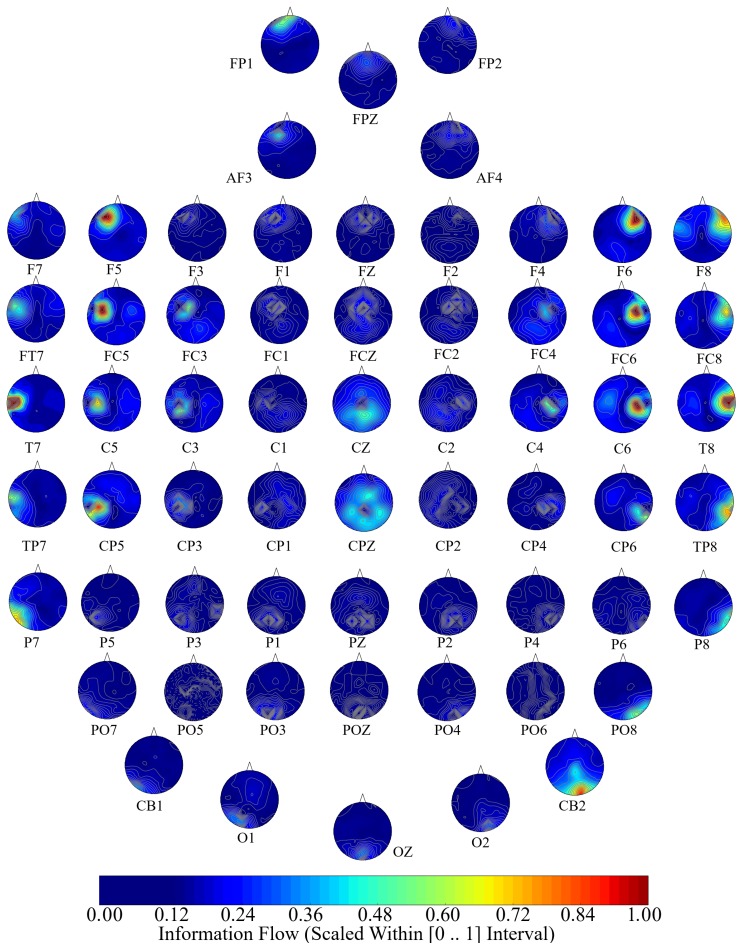
Neutral affect’s channel-wise information flow. These subplots identify a bi-hemispheric brain activity in response to Neutral affect. They also show that a number of channels are associated with higher short- as well as long-range information (e.g., F6, F8, FC5, FC6, CZ, CP5, T8, CB2). Although these channels appear to have higher local influence in the form of information flow, their corresponding flow of information extend beyond their designated hemispheres, thereby indicating the presence of cross-hemispheric whole-brain information flow and communication. The values in these subplots are scaled within [0,⋯,1] for better comparison.

**Figure 7 brainsci-10-00008-f007:**
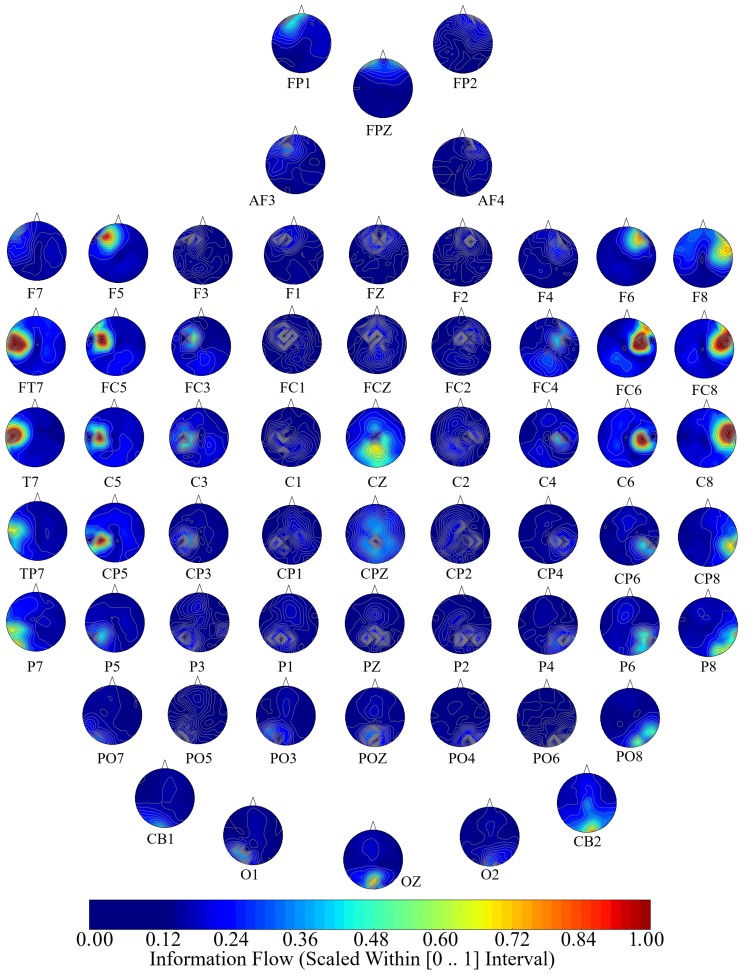
Positive affect’s channel-wise information flow. These subplots identify a bi-hemispheric brain activity in response to a Positive affect. They also show that a number of channels are associated with higher short- as well as long-range information (e.g., F8, FT7, FC5, FC6, FC8, C5, CZ, C6, CP5, CB2). Although these channels appear to have higher local influence in the form of information flow, their corresponding flow of information extend beyond their designated hemispheres, thereby indicating the presence of cross-hemispheric whole-brain information flow and communication. The values in these subplots are scaled within [0,⋯,1] for better comparison.

**Figure 8 brainsci-10-00008-f008:**
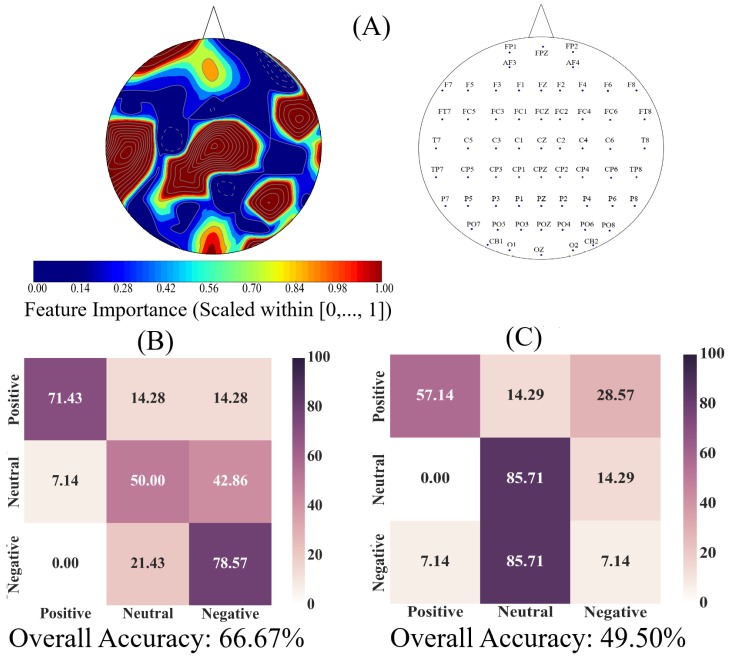
(**A**) spatial map of feature importance by Adaboost meta-estimator pertinent to the trained Adaboost meta-estimator on the ucd values associated with Negative, Neutral, and Positive affects (one-sample bootstrap test of significance at 95% confidence interval: M = 0.13, SD = 0.03, CI = [0.08 0.21]). The right subplot shows the EEG channels’ arrangement; (**B**) Adaboost prediction accuracy in Negative, Neutral, and Positive affects in 1-holdout setting using whole-brain ucd values; (**C**) Adaboost prediction accuracy in Negative, Neutral, and Positive affects in 1-holdout setting using subset of channels with their importance within or above the one-sample bootstrap test of significance (10,000 simulation runs) at 95.0% confidence interval on these feature importance values. In (**B**,**C**), the correct predictions, per affect, are the diagonal entries of these tables and the off-diagonal entries show the percentage of each of the affects that was misclassified (e.g., Positive affect misclassified as Negative affect).

**Table 1 brainsci-10-00008-t001:** Bootstrap (10,000 simulation runs) 95.0% confidence intervals (CI) associated with the Spearman correlation between Negative, Neutral, and Positive affects.

Conditions	*r*	*p* (Two-Tailed)	CI${}_{95\%}$
Positive vs. Negative	0.58	0.00001	[0.53 0.63]
ine Positive vs. Neutral	0.58	0.00001	[0.53 0.63]
ine Negative vs. Neutral	0.53	0.00001	[0.47 0.59]

**Table 2 brainsci-10-00008-t002:** Paired two-sample bootstrap test of significance (10,000 simulation runs) at 95.0% confidence interval (CI) associated with the participants’ whole-brain unit causal density (ucd) values. Compared pairs of affect are: Positive versus Neutral, Positive versus Negative, and Negative versus Neutral. M and SD refer to the mean difference and the standard deviation of such a difference between the two compared states. CI shows the 95% confidence interval of their difference. Bold entry rows indicate the significant difference.

Conditions	M${}_{\mathrm{difference}}$	SD${}_{\mathrm{difference}}$	95.0% CI${}_{\mathrm{difference}}$
**Positive versus Neutral**	**0.07**	**0.02**	**[0.02 0.11]**
ine **Positive versus Negative**	**0.09**	**0.02**	**[0.04 0.14]**
ine Negative versus Neutral	−0.02	0.02	[−0.07 0.02]

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
