# Peer review of "Emergence of the Affect from the Variation in the Whole-Brain Flow of Information"

_brainsci, 2019, doi:10.3390/brainsci10010008_

Round 1

Reviewer 1 Report

This manuscript details the results of a study examining whether the brain regional interactivity can provide further insights for understanding the neural substrates of the affect. Through Granger causal analysis, the authors explored the EEG recordings in 15 human subjects during a task with movie clips supposed to elicited positive, negative and neutral affect states. The authors suggest that the affect emerge from the variation in the information flow among distributed brain regions, located in both hemispheres. Despite the fact that this topic constitute an interesting research area, I have several major concerns about this manuscript, both in form and substance.

The objectives of the study are not clearly established, neither in the abstract nor in the introduction.

The introduction is very long, but nevertheless, we do not really understand the main interest of this study. What are the objectives? What are the shortcomings of the current literature? Why is it important to study these concepts and what can it bring new? What are the hypotheses of the study? 

The construction of the article is not flowing, the methodology comes after the results and the discussion (non-sense).

There’s a lot of figures, and some of them should be in Supplementary material because it’s too ponderous and also because the authors don’t really discuss them.

The methodological part is really incomplete: - There’s only few details about the task : Is this task validated ? Pretests ? What kind of stimulus ? With or without sound ? etc. - There’s only few details about the sample : what kind of population? Education ? Inclusion criteria ? - Also, a justification of the method used is needed : Why EEG (versus other technique such as MRI etc) ? A more detailed description of the basics of EEG method for non-specialist.

The sample is too small. 15 healthy young subjects for this kind of study is small to make conclusion on the human brain, especially with techniques like EEG. The size of the sample should be increased in order to use parametric, more powerful and reliable statistics.

Finally, the manuscript should be reworked for the English, the sentences are very long, very difficult to understand, and sometimes it induces misinterpretations.

In conclusion, this manuscript would need to be more detailed on some points, but lightened on others. The analyzes should be better justified, and carried out on a larger sample of subjects. An abstract / introduction and a discussion more straight to the point would be appreciated.

Author Response

First and foremost, the authors would like to take this opportunity to thank the reviewers and the associate editor for their time and kind consideration to review our manuscript. The comments by the reviewers enabled us to improve the quality of our results and their presentation substantially.

In what follows, we provide point-by-point responses to the comments and concerns raised by the reviewers 1.

Sincerely,

Reviewer 1

Reviewer’s Comment: The objectives of the study are not clearly established, neither in the abstract nor in the introduction.

Authors’ Response: In the first version of our manuscript, we summarized (Section 1. Introduction, lines 41-51,in the current version of the manuscript) the main three hypotheses about the neural substrates of the affect and also highlighted the compelling and yet contrasting results among their related studies. We next attributed (Section 1. Introduction, lines 52-61,in the current version of the manuscript) this to two primary reasons that our overview of the literature appeared to be suggestive of.

However, the reviewer’s comment made it clear thatwe did not provide sufficient insight on the purpose of our study. Therefore, we applied the following changes to the current version of the manuscript.

Section 1. Introduction

Lines 86-100:We underlined the shortcomingsof the current studies about the brain regional interactivity. We also explained that how such approaches as Granger causality can help address these shortcomings. We then specifically pointed out how our approach can benefit the study of affect. The new modification reads as follows.

“Although previous research aimed at identifying the functional interaction among brain regions, it mostly framed such an interactivity in terms of statistical association (e.g., correlation) [63]. However, this approach to the study of brain regional interactivity is problematic since such associations as correlation can arise in a variety of ways that do not entail causal relation (i.e., directional flow of information) [63]. As a result, they do not allow for understanding the mapping between such associations and their underlying neural substrates [63,65]. Addressing these shortcomings can be facilitated by utilization of such approaches as Granger causality that provide means to establish directional relations between the brain regions. For instance, it can help verify whether the observed associations were indeed stemmed from causal (i.e., in its purely statistical term) relations among these regions. Furthermore, it can enable researchers to investigate the existing theorems and hypotheses about the brain functioning and its regional interactivity in a more robust way. For example, G-causality can be adapted for the study of the affect in terms of brain’s regional functional connectivity, thereby allowing for reconciliation among the contrasting results of the bipolarity [19], bivalent [2022], and affective workspace [18,23] hypotheses [18,2731]. To the best of our knowledge, no previous study has considered the use of Granger causality for this purpose.

Lines101-112:We then updated our Contributions paragraph by adding Taken together, our findings appear to be more in line with the affectiveworkspace hypothesis [18,23] than the bipolarity [19] or the bivalent hypotheses [2022].”The full paragraph reads as follows.

“Our contributions are threefold. First, we show that the Negative, Neutral, and Positive affect emerge from subtle variation in information flow of the brain regions that are in both hemispheres. Second, we show that these regions are common between these affect states than distinct to a specific affect. Third, we show that these regions are characterized with both short- as well as long-range information flow. This provides evidence for the presence of simultaneous integration and differentiation in the brain functioning that leads to the emergence of different affect. Taken together, our findings appear to be more in line with the affective workspace hypothesis [18,23] than the bipolarity [19] or the bivalent hypotheses [2022]. Our results are also in line with the findings on the presence of intrinsic large- scale interacting brain networks that underlie the production of psychological events [7275]. We believe that our study can help advance our understanding of the neural basis of the human’s emotions by identifying the signatures of differential affect in subtle variation that occurs in the whole-brain cortical flow of information.”

Section 4. Discussion

Lines 493-503:We elaborated on why we considered our results to be more in line affective workspace hypothesis [18,23]. It reads as follows.

Our findings appeared to be more in line with the affective workspace hypothesis [18,23] than the bipolarity [19] or the bivalent hypotheses [2022]. For instance, the bipolarity hypothesis [19] attributes the Positive and Negative affect to the opposing ends of a single dimension [24,25]. As a result, it is plausible to expect anti-correlations between these affect states’ flow of information. Contrary to this expectation, we observed positive correlations among them. On the other hand, the bivalent hypothesis [2022] emphasizes on the presence of two distinct and independent brain systems for the Positive and Negative affect [2022]. This makes it plausible to expect that the information flow associated with these affect states to form non-overlapping and distinct patterns. Although our analyses identified a number of brain regions whose unit causal flows significantly differed between Negative, Neutral, and Positive affect (see Appendix B), we also observed that these regions were common between these affect.

Lines504-522:We provided further support with regard to above as well as our stance on emergence of the affect from variation in the cortical flow of information using the results of linear classifier. It reads as follows.

Further evidence in support of this view came from the classification results of these affect states in which a simple linear model was sufficient to significantly distinguish between these affect states. These results were in line with our correlation analyses in that they implied that any change in the information flow among whole-brain distributed cortical regions in one affect can be explained in terms of a linear change in the corresponding flow of information in the other affect state that was in the same direction as of the first one. We also observed that a linear model that solely utilized the regions with significantly different flow of information only, yielded a significantly poorer performance in comparison with the setting in which the whole-brain cortical information flow was considered. Additionally, the mere use of these regions appeared to cause the linear model to treat all three affects as Neutral. This interpretation can be verified by observing the sudden and substantial increase in linear model’s correct classification of the Neutral affect that was accompanied by its significantly reduced accuracy in the case of both Positive and Negative affect states. In fact the recurrence of some of these regions in such domains as social cognition and theory of mind [97], story comprehension [91,92], autobiographical memory [98], decision making [99], working memory [100], and self-referential processing [101], along with the reduced accuracy of the linear model in our study further implied that [1] these regions may not be specific to the affect but may constitute to other cognitive and perceptual events, thereby forming domain-general networks [102,103]. Considering these observations, our findings indicated that the whole-brain flow of information that was manifested through distributed cortical regions were not only shared among the Negative, Neutral, and Positive affect states but also was able to best distinguish between these three affect states.

NewSection5. Concluding Remarks:Itadded the following contents.

Lines 568-569:The first paragraph summarized the overall findings and brought them main manuscript’s results and the Appendices together. In particular, it discussed how the causal density and causal flows provided some supports for the possibility of the emergence of the affect from variation in the whole-brain cortical flow of information. It reads as follows.

In this article, we provided evidence for the possibility of the emergence of the affect from

the variation in the information flow among distributed brain regions that were located in both hemispheres. We supported this viewpoint by showing that the whole-brain cortical flow of information was positively correlated between these affect states, that although these distributed regions were shared among the Negative, Neutral, and Positive affect, they appeared to share information differentially in response to these affect, and that a simple linear model was able to distinguish between these affect states with a significantly above average accuracy when it used the whole-brain cortical flow of information. For instance, whereas we observed a non-significant difference between the channels’ causal densities in the case of Negative and Neutral affect, we found a considerable number of channels’ whose causal flows differed significantly in response to these two affect states. These results hinted at the possibility of interpreting the cortical responses to these affect in terms of the variation in the flow of information that differed significantly among these channels. In other words, the observed differences suggested that although these channels were common between the Negative and the Neutral affect and that their causal densities (i.e., the overall degree of causal interactivity) were non-significant, their causal flows (i.e., the extent of their influence on/by the other channels) were significantly varying in response to these differential affect.

Lines 568-600:Next, we discussed the interesting and puzzling results with respect to the classifier’s performance that showed more misclassification of the Neutral as Negative than Positive affect. It reads as follows.

In this article, we provided evidence for the possibility of the emergence of the affect from

the variation in the information flow among distributed brain regions that were located in both hemispheres. We supported this viewpoint by showing that the whole-brain cortical flow of information was positively correlated between these affect states, that although these distributed regions were shared among the Negative, Neutral, and Positive affect, they appeared to share information differentially in response to these affect, and that a simple linear model was able to distinguish between these affect states with a significantly above average accuracy when it used the whole-brain cortical flow of information. For instance, whereas we observed a non-significant difference between the channels’ causal densities in the case of Negative and Neutral affect, we found a considerable number of channels’ whose causal flows differed significantly in response to these two affect states. These results hinted at the possibility of interpreting the cortical responses to these affect in terms of the variation in the flow of information that differed significantly among these channels. In other words, the observed differences suggested that although these channels were common between the Negative and the Neutral affect and that their causal densities (i.e., the overall degree of causal interactivity) were non-significant, their causal flows (i.e., the extent of their influence on/by the other channels) were significantly varying in response to these differential affect.

On the other hand, a puzzling observation with regards to the linear classifier was the higher percentage of misclassification between the Neutral and the Negative affect. We noticed that this could not readily be attributed to the sample size since we used a balanced dataset in our analyses (i.e., equal number of samples for each of the Negative, Neutral, and Positive affect, per participant). Another possible reason behind the observed effect could have been the difference in the shared information between these affect. However, the results of the correlation analyses (i.e., linear measure of mutual information [121]) did not provide any further insight on this matter. For instance, these results could not explain the higher misclassification of the Neutral as the Negative than the Positive affect, despite the fact that the Neutral affect appeared to share slightly higher information (i.e., comparably higher correlation) with the Positive than the Negative affect. Furthermore, the observed correlations between Positive and the other two affect states were equivalent. As a result, if the higher shared information was to blame, then it should have caused a comparable misclassification between the Positive and the Negative which evidently was not the case. Conversely, the lower shared information also was not sufficient to account for the observed higher misclassification of the Neutral affect as the Negative than the Positive affect. This is due to the observation that the high rate of misclassification between the Neutral and Negative should have in principle been the lowest, given their lowest correlation among the three pairwise comparisons.

Lines601-610:We then discussed the implications of the above two paragraphs along with the needs for and potential direction of the future research with regards our observations and results. It reads as follows.

The implications of these observations were threefold. First, although the Negative, Neutral, and Positive affect appeared to share a common distributed neural correlates, the underlying dynamics of such a whole-brain cortical information flow might be shared differentially among them. Second, such a dynamics might have more in common between the Negative and the Neutral affect states than the Neutral and the Positive or the Negative and the Positive affect. Third, this potential dynamics that governed the cortical responses to these affect might not primarily be explained through a linear modeling of such a common and distributed neural activity. Therefore, future research that takes into account the non-linear dynamics of neural activity [54,117] in response to Negative, Neutral, and Positive affect is necessary to further extend our understanding of the potential causes of the observed dis/similarities between these affect states.

Lines611-620:Wealsoemphasizedthe necessity for a larger and more diverse sample size for drawing more informed conclusion on the emergence of the affect. It reads as follows.

“Many neuroscientific studies are based on a small number ofparticipants [118121]. For instance, the recent comprehensive meta-analysis by Lindquist [18] that reported on an extensive coverage of 914 experimental contrasts in affect-related studies accounted for 6827 participants (i.e., 6827/914 ≈ 7.47 participants on average). Furthermore, most of the individuals that are included in these studies share the same geographical and/or cultural background (but also see [94] for a small deviation). Neuropsychological findings indicate that the individuals’ ability to experience pleasant/unpleasant feelings to express these subjective mental states in terms of such attributes as positive or negative to be the unifying and common concept across cultures [8,9].However,future research that includes a larger human sample as well as different age groups andmore cultural diversity is necessary for drawing a more informed conclusion on the findings that were presented in this article.”

With regards to two limitations that we highlighted in Section 1. Introduction, lines 52-61,we also referred to them in the Abstract, lines 5-10,in the current version of the manuscript. We then modified the following two sentences of the Abstract (lines 10-14, in the current version of the manuscript) to summarize how we addressed them using Granger causality. These two sentences read as follow.

To address these limitations, we performed Granger causal analysis on the EEG recordings of the human subjects who watched movie clips that elicited Negative, Neutral, and Positive affect. This allowed us to look beyond the brain regional activation in isolation to investigate whether the brain regional interactivity can provide further insights for understanding the neural substrates of the affect.”

Reviewer’s Comment: The introduction is very long, but nevertheless, we do not really understand the main interest of this study. What are the objectives? What are the shortcomings of the current literature? Why is it important to study these concepts and what can it bring new? What are the hypotheses of the study?

Authors’ Response: Please refer to our response to “Reviewer’s Comment: The objectives of the study are not clearly established...”above.

Reviewer’s Comment: The construction of the article is not flowing, the methodology comes after the results and the discussion (non-sense).

Authors’ Response: We apologize for the inconvenience that this matter may have caused the reviewer. Although we used an MDPI journal’s template, we chose the wrong one (i.e., MDPI Journal Entropy) that instructs the authors for having the format we adapted in our initial submission. We addressed this issue in the current version of the manuscript in which the Materials and Methods proceeds the Results and Discussion Sections.

Reviewer’s Comment: There’s a lot of figures, and some of them should be in Supplementary material because it’s too ponderous and also because the authors don’t really discuss them.

Authors’ Response: We moved the materials that were associated with the paired channel-wise significant differences between Negative,Neutral, and Positive affect to Appendix B Channel-wise Comparison of the Information Flow in Negative, Neutral, and Positive Affect (lines642-655, along with Figures A2-A4, and Tables A2-A4).The content of this new Appendix reads as follows.

Only some of the observed significant differences based on channel-wise paired Wilcoxon rank sum test passed the posthoc two-sample bootstrap test of significance (10,000 simulation runs) at 95.0% confidence interval. In the case of positive versus neutral (FigureA2 and Table A2) whereas this test identified CPZ with the higher information flow in neutral than positive, it indicated FT7, FT8, P5, P1, P4, P6, PO5, POZ, PO6, and OZ to have higher flow of information in positive than the neutral. In the case of positive versus negative (Figure A3and Table A3), we observed that whereas negative affect was associated with higher information flow in F3, positive affect showed significantly higher information flow than negative affect in FP1, F7, F2, T8, CP2, P3, P1, P2, P4, PO5, PO8, and OZ. Finally, for negative versus neutral (Figure A4and Table A4), the neutral affect was associated with the higher information flow than negative in the case of F1, F4, FCZ, C4, and CP2 while negative affect had higher flow of information than neutral in F3, FT7, FT8, P5, P4, P6, PO5, POZ, PO6, and O2.

Taken together, the two-sample bootstrap test of significance (10,000 simulation runs) at 95.0% confidence interval identified the relation positive > negative > neutral in channels P4 and PO5 and the relation positive & negative > neutral in channels FT7, FT8, P5, P4, P6, PO5, POZ, and PO6.”

We also added the following line to Section 3.3. Unit Information Flow (lines 423-425)to reflect these changes.

Further details on channel-wise comparison of the information flow in negative, neutral, and positive affect are presented in Appendix B.”

Reviewer’s Comment: The methodological part is really incomplete: - There’s only few details about the task : Is this task validated ? Pretests ? What kind of stimulus ? With or without sound ? etc. - There’s only few details about the sample : what kind of population? Education ? Inclusion criteria ? - Also, a justification of the method used is needed : Why EEG (versus other technique such as MRI etc) ? A more detailed description of the basics of EEG method for non-specialist.

Authors’ Response: We outlinedthe changes and modifications that we applied to address the reviewer’s concern in the following threesteps: “Missing details about the experiment,”Justification of the use of EEG,” and “Description of EEG neuroimaging”.

Missing details about the experiment: We rewrote the Section 2.1. The Dataset (lines124-168,in the current version of the manuscript)and also the Figure 1’s caption (page 3, in the current version of the manuscript).Specifically, we included information about task’s validation, pretest steps, type of stimulus (i.e., audiovisual movie clips),information about the participants. In the current version of the manuscript,the modified Section 2.1. The Dataset (lines 94-136)reads as follows.

“SEED [66] corresponds to sixty-two-channel EEG recordings (Figure 1(B)) of fifteen Chinese subjects (7 males and 8 females; Mean (M) = 23.27, Standard Deviation (SD) = 2.37). All participants were right-handed (with self-reported normal or corrected-to-normal vision and normal hearing) and were students from Shanghai Jiao Tong University. They watched fifteen Chinese movie clips (four minutes in duration) that elicited negative, neutral, and positive affect. These individuals were selected based on the Eysenck Personality Questionnaire (EPQ) [72] personality traits. EPQ evaluates the individuals’ personality along three independent dimensions of temperament: Extraversion/Introversion, Neuroticism/ Stability, and Psychoticism/Socialisation. Eysenck et al. [72] reported that it appears that not every individual can elicit specific emotions immediately (even in the presence of explicit stimuli). They also reported that individuals who are extraverted and have stable moods tend to elicit the right emotions throughout the emotion-based experiments. Therefore, the authors in SEEDadapted the same personality criteria that was reported by Eysenck et al. [72] to select the fifteen individuals that participated in their experiment.

Prior to SEED experiment, the authors asked twenty volunteers to assess a pool of movie clips in a five-point scale. Based on the result of this assessment, they selected the fifteen movie clips (i.e., five clips per Negative, Neutral, and Positive affect) whose average score were ≥ 3 and ranked in the top 5 in each affect category. They further verified that the selected movie clips indeed elicited the targeted affect in a follow-up study [82] that included nine separate individuals who were different from the twenty volunteers that originally involved in rating and selection process of fifteen movie clips. The authors then used these movie clips in SEED experiment.

In SEED, each experiment included (Figure 1(A)) a total of fifteen movie clips, per participant. In this setting, each movie clip was proceeded with a five-second hint to prepare the participants for its start. This was then followed by a four-minute movie clip. At the end of each movie clip, the participants were asked to answer three questions that followed the Philippot [74]. These questions were the type of emotion that the participants actually felt while watching the movie clips, whether they previously watched the original movies from which the clips were taken, and whether they understood the content of those clips. The participants responded to these three questions by scoring them in the scale of 1 to 5. The participants were then instructed to take a fifteen-second rest before the next movie clip in the experiment started. Each individual participated in three experiments with an interval of about one week between them. The same set of fifteen movie clips were used in all of these three experiments. Every participant watched the same set of fifteen movie clips in the same order of their presentations.

The movie clips within each experiment were ordered in such a way that two clips with the same emotional content (e.g., both targeting negative affect) were not presented consecutively to the participants. Additionally, these clips were selected based on the criteria that their lengths were not too long to induce fatigue on the subjects while watching them, that their contents were easy to understand by the participants without any explicit explanation, and that each clip elicited a single desired target affect (e.g., negative or positive).

SEED comes with its preprocessed EEG recordings which we used in the present study. Its preprocessing steps consist of downsampling the EEG recordings to 200 Hz followed by bandpass filtering the signals within 0-75 Hz. These steps were applied on the extracted EEG segments that corresponded to the duration of each movie clip. Further details on SEED experiment, EEG channels’ arrangement, movie clips’ selection criteria, data acquisition and preprocessing, labeling the emotional states associated with each movie clip, etc. can be found in [66]and http://bcmi.sjtu.edu.cn/ seed/seed.html.”

We also updated the caption of the figure that was associated with the experimental setting (i.e., Figure 1 (A), page 4,in the current version of the manuscript) as follow.

Figure 1. (A) Schematic diagram of an experiment as described in [66]. Each experiment included a total of fifteen movie clips (i.e., n = 15, audiovisual), per participant. In this setting, each movie clip was proceeded with a five-second hint to prepare the participants for its start. This was then followed by a four-minute movie clip. At the end of each movie clip, the participants were asked to answer three questions that followed the Philippot [74]. These questions were the type of emotion that the participants actually felt while watching the movie clips, whether they watched the original movies from which the clips were taken, and whether they understood the content of those clips. The participants responded to these three questions by scoring them in the scale of 1 to 5.

Justification of the use of EEG: We added the following paragraph to Section 2.4. Statistical Analysis (lines239-246,in the current version of the manuscript).

The EEG time series that were considered in this study had a relatively large sampling rate (i.e., 200 Hz after downsampling) and corresponded to a four-minute experimental setting. As a results, they were long enough to allow for the choice of such dynamical analysis as Granger causality, given the sensitivity of such techniques to the length of time series. In addition, EEG has a higher temporal resolution (i.e., in comparison with other neuroimaging techniques such as fMRI and fNIRS) and therefore poses itself as a better choice for the study of causal connectivity (i.e., in a pure statistical than anatomical sense). Detailed discussion on such issues and considerations can be found in [52,53,57,58,66,67].”

Description of EEG neuroiamging for non-specialist:We added the following paragraph to Section 1.Introduction(lines80-83,in the current version of the manuscript).

“EEG is an electrophysiological monitoring method that records the brain’s spontaneous electrical activity over a period of time. These electrical activities are due to the voltage fluctuations induced by ionic current within the neurons [67]. EEG recordings are generally acquired through multiple electrodes.

Reviewer’s Comment: The sample is too small. 15 healthy young subjects for this kind of study is small to make conclusion on the human brain, especially with techniques like EEG. The size of the sample should be increased in order to use parametric, more powerful and reliable statistics.

Authors’ Response: This is indeed an important issue that is quite pervasive in neuroscientific studies which can be due to the difficulty associated with conducting such studies (e.g., tedious and time-consuming procedure for recruiting human subjects that best suites the experimental paradigm, additional documentation and permission requirement associated with studies that involve the use of human subjects and other animals, finding the targeted age-range and/or gender-balance, etc.). In this respect,many studies draw their observations using a small number of participants. For instance, [94] included 18 participants who watched series of emotional images. Similarly, in [95] 18 participants watched a 25-minute movie clip, in [96] 13 participants watched colored images,and [97] included 9 participants only.Furthermore, many of these studies rely on individuals who mostly share the same geographical/cultural background. As an exception, our overview showed that [89] included 15 participants, three of whom were discarded due to the issues related to technical reasons/poor data quality (i.e., in total 12 participants were included in [89]). Out of these 12 participants, 11 were native Finnish speaker and 1 only was a native English speaker.

In fact, the comprehensive meta-analysis that was published in 2016 about the neural correlates of the affect (i.e., reference [25] in the current version of the manuscript) included 914 experimental contrasts that accounted for 6827 participants over the period 1993-2011 (i.e., 6827/914 ≈ 7.47 participants on average).

However, we also appreciateand agree with the reviewer’scomment on the importance of this issue. Therefore, we addeda new Section 5. Concluding Remarksto the current version of the manuscript in which we discussed this issue and the need for future study using larger and more diverse sample of human subjects which reads as follows (lines611-620).

Many neuroscientific studies are based on a small number ofparticipants [9497]. For instance, the recent comprehensive meta-analysis by Lindquist [25] that reported on an extensive coverage of 914 experimental contrasts in affect-related studies accounted for 6827 participants (i.e., 6827/914 ≈ 7.47 participants on average). Furthermore, most of the individuals that are included in these studies share the same geographical and/or cultural background (but also see [89] for a small deviation). Neuropsychological findings indicate that the individuals’ ability to experience pleasant/unpleasant feelings to express these subjective mental states in terms of such attributes as positive or negative to be the unifying and common concept across cultures [11,12].However,future research that includes a larger human sample as well as different age groups along with more cultural diversity is necessary for drawing a more informed conclusion on the findings that were presented in this article.”

With regards to our choice to opt for non-parametric tests, we checked the participants’ unit causal density and the unit causal flow in each of the negative, neutral, and positive affect states (separately as well as combined, with respect to the both individuals and the EEG channels for each of the affect)and found that they did not follow the normal distribution. Therefore, we chose non-parametric tests. We also observed that our analyses of the participants’ unit causal densitiesyielded small effect sizes (Section 3.2. Unit Causal Density, lines 368-374, in the current version of the manuscript).Therefore, we decided to further apply the bootstrap tests,therebyallowingus to verify the validity of observed significant differences. Specifically, by adapting thistest (i.e., random sampling with replacement) that was carried for 10,000 simulation runs (therefore allowing for potential outliers to be repeated with higher probability and more frequently) at 95% confidence interval (CI) (i.e., p < .05 significance level) enabled us to observe whether these results were indeed reliable, given the small sample in the present study: the CI made it possible to verify thatour results exhibited the central tendency and the overall test showed thatthedistribution of our test results approximated the normal distribution, thereby adhering the law of large number and the central limit theorem. To clarify this, we added the following to the current version of the manuscript (Section2.4.3. Importance of Channels’ Unit Causal Densities, lines 338-352).

With regard to our analyses, there are two points that are worth further clarification. They are the choice of non-parametric tests and the follow-up bootstrap test of significance. Prior to our analyses, we checked the participants' unit causal density and the unit causal flow in each of the negative, neutral, and positive affect states (separately as well as combined, with respect to the both individuals and the EEG channels for each of the affect) and found that they did not follow normal distribution. Therefore, we opted for non-parametric analyses. In the case of bootstrap, on the other hand, we realized that our analyses were performed based on a small sample of participants (i.e., fourteen individuals). We also observed that our analyses of the participants’ unit causal flowyielded small effect sizes. Therefore, it was crucial to ensure that any significant results that we observed in our analyses were not due to a subsample of individuals (i.e., distorted data and hence lack of central tendency). This concern was further strengthened by the result of the non-normality of the participants' unit causal density and the unit causal flow. Therefore, we decided to also apply the bootstrap test (i.e., random sampling with replacement) that was carried out for 10,000 simulation runs (therefore allowing for potential outliers to be repeated with higher probability and more frequently) at 95% confidence interval (i.e., p < .05 significance level), thereby enabling us to further verify our results.

Reviewer’s Comment: Finally, the manuscript should be reworked for the English, the sentences are very long, very difficult to understand, and sometimes it induces misinterpretations.

Authors’ Response: We further audit our manuscript and checked for the concerns that were raised by the reviewer. In the current version of the manuscript, we paid special attention to break the long sentences into two or three sentences to increase the comprehensibility of the content of the manuscript. We also ensured our manuscript conveyed a coherent message throughout and subsequently modified any sentence that we found might have been a source of misunderstanding and/or misinterpretation. However, we would also like to ask the reviewer to further let us know of any part(s)that the reviewer found specifically unclear and we might stillhave missed.

Reviewer 2 Report

The paper is a sound study of emotional dynamics in the neocortex. It claims to examine ``the possibility of the emergence of the affect from the variation in the information flow among distributed brain regions that were located in both hemispheres." However in fact affect emerges at a pre-cortical level as discussed inter alia by Panksepp, influencing cortical activity via  neuromodulators such as dopamine spread through the neo-cortex via the ascending systems. This study does not have the capacity to detect such pre-cortical activity, which is the key to actually discussing the emergence of affect, rather than the cortical reponse to affect, which is what this paper is about. 

Its overall an excellent study insofar as what it does, but should add a paragraph to comment on  these subcortical aspects of affect, which occur through deciated physiological structures (the "ascending systems"). This would properly situate the study in a larger picture (after all it is well known that these neuromodulators play a key role in emotional dynamics, and evolution has led to their existence because of their importance)

Author Response

First and foremost, the authors would like to take this opportunity to thank the reviewers and the associate editor for their time and kind consideration to review our manuscript. The comments by the reviewers enabled us to improve the quality of our results and their presentation substantially.

In what follows, we provide point-by-point responses to the comments and concerns raised by the reviewer 2.

Sincerely,

Reviewer 2

Reviewer’s Comment: The paper is a sound study of emotional dynamics in the neocortex. It claims to examine ``the possibility of the emergence of the affect from the variation in the information flow among distributed brain regions that were located in both hemispheres." However in fact affect emerges at a pre-cortical level as discussed inter alia by Panksepp, influencing cortical activity via neuromodulators such as dopamine spread through the neo-cortex via the ascending systems. This study does not have the capacity to detect such pre-cortical activity, which is the key to actually discussing the emergence of affect, rather than the cortical reponse to affect, which is what this paper is about.

Its overall an excellent study insofar as what it does, but should add a paragraph to comment on these subcortical aspects of affect, which occur through deciated physiological structures (the "ascending systems"). This would properly situate the study in a larger picture (after all it is well known that these neuromodulators play a key role in emotional dynamics, and evolution has led to their existence because of their importance)

Authors’ Response: This indeed an important point raised by the reviewer that authors missed to emphasized on.We addressed the reviewer’s concerns in two parts of our manuscript.

Section 1. Introduction, lines 113-121:We added the following paragraph to the end of this Section to clarify that the affect originates from subcortical firing and neuromodulatory mechanisms.

In regards to our study, there is an important point that deserves further clarification. A number of neuroscientific findings argue that the basic emotions are localized to the firing of subcortical circuits [76,77]. They also show that the emergence of the affect rather originates from these subcortical regions [77] where multiple brainstem-derived modulatory neurotransmitters contribute to emotion and emotional behaviour [7880]. In this respect, it is crucial to note that the present study is not about the origin of the affect in such subcortical levels. It primarily aims at the higher-level cortical regions to verify whether these regions are common/distinct to/between the Negative, Neutral, and Positive affect. Subsequently, it investigates the extent to which the potentially differential flow of information among these regions can account for neural substrates of the Negative, Neutral, and Positive affect.”

Section5. Concluding Remarks, lines 621-626:We added the following paragraph to emphasize the necessity to extend our results to the cases in which cortical and subcortical activities are acquired simultaneously, thereby allowing to draw the correspondence between the observed cortical-level information flow and sharing and their underpinning subcortical mechanisms and dynamics.

“Last, it would also be interesting to extend the results in the present study to the case in which the cortical activities are simultaneously acquired alongside the brain subcortical responses to the affect. [76,77]. Such studies can shed light on the underlying mechanisms that lead to the observed distributed variation in the whole-brain cortical flow of information, thereby allowing for the better understanding of the correspondence between these cortical information sharing and their underpinning neuromodulatory mechanisms and dynamics [7880].

We also tried our best to further indicate this throughout the manuscript by adding the term “cortical” to occurrences of the phrase “whole-brain flow of information” and/or “whole-brain information flow”. Some examples include:

SectionAbstract, line 23

SectionIntroduction, line 112

SectionDiscussion, lines458, 510, and 566

SectionConcludingRemarks, lines 570, 574, 603, and 624

Reviewer 3 Report

This work presents a characterization of the brain interactions estimated from recordings done under different affect states, i.e. positive, negative and neutral, in terms of Granger causality. Authors show that a subset of channels differently interacts among them depending on the affect states. Moreover, it is also proved the capability of a decoder to predict the affect state from the causality measure between each EEG channel pair.

Overall, I think that the work is interesting and provides important results in the neuroscience community. But anyway I do not consider the manuscript ready to be published at the current stage.

Firstly, I would suggest to structure the manuscript according to the instruction here: https://www.mdpi.com/journal/brainsci/instructions#manuscript. Placing the result section immediately after the introduction makes the understanding of the work difficult. Moreover, I suggest to give a precise definition of the measures called ucd and unit information flow (and sometimes causal flow) before the result section.

Regarding the results in subsection 2.3, in particular, Table 3/4/5 I wonder about the possibility to apply a repeated measure ANOVA for each channel with factor the affect states instead of exploring each pair of affect states.

In my opinion, the most interesting result is the significant performance of the decoder in the affect state classification but actually authors do not give high relevance to it. Even the title of 2.4 focuses on the channel importance that to me it is like a side result of the classification. I mean firstly I would care about the capability of discriminating between classes and then I would wonder about the feature importance.

An overall comment, there are a lot of inaccuracies in the result presentation. Basic information related to the plotted results is missing. Moreover, figure quality should be improved, e.g. axis labels sometimes are repeated even if more plots share the same labels, while there are cases in which labels are written only in the first subplot (see the first column vs the second column of subplots in Figure 1)

Some specific comments:

Introduction line 58: Authors introduced the concept of Granger causality in the framework of the "nonlinear dynamical system analysis". I think this statement should be better explained because the most common implementation of GC is based on the linear AR model, and this is also the case for the Barnett and Seth toolbox. It is not clear to me what type of nonlinearity you are referring to.

2.1 Correlation:
- ucd is not defined, it is strange that the next subsection (2.2) titles Unit Causal Density while this measure has been already used before without giving a meaning to it;
- in Figure 1 part A, B and C are not specified;
- it is not clear why 95%CI is [.68,.7] given that the SD=.3;
- I would suggest to report the % of participants with an ucd value < significant level;
- 'see A' at the end of the figure explanation, I suppose means Appendix A;

2.3 Unit Information Flow:
- parenthesis line 153;
- description of Table 4, error in Negative>Neutral
- sometimes negative, positive and neutral start with a capital letter other times no, there should be consistency;
- figure 4/5/6, according to the colorbar it seems that there are values (blue levels) lower than 0;
- figure 7/8/9, figure quality should be improved, i.e. more consistency in the choice of the x and y scale, axis labels not in the last subplot but at the global level;
- table 2/6/7/8 I would suggest to highlight somehow, e.g. bold character, significant channels;
- table 6 and figure 7 convey the same type of information (as also the other pairs of table/figure), my opinion is that it is redundant to show both, maybe the histograms can be shown in the appendix;
- table 6, there is a separation between channels where neutral>positive and vice-versa, while in table 7 and 8 not;

Looking at Table 8, I was surprised noticing that most of the channels are significant while from Figure2C there is no significant difference between negative and neutral. Is there an explanation or am I missing something?

2.4 Importance of channels' unit causal densities
- colorbar of figure 10A is not clear, according to the text description values are scaled in [0,1] while in the colorbar negative values are included and the maximum is 0.84. Moreover, the colors are not uniformly distributed.

4.1
- Figure 11 "secends"

4.3
- A formal definition of unit causal density and causal information flow should be given. The actual description is confusing. I suggest to use a mathematical formula instead of phrasing the concept;

Author Response

First and foremost, the authors would like to take this opportunity to thank the reviewers and the associate editor for their time and kind consideration to review our manuscript. The comments by the reviewers enabled us to improve the quality of our results and their presentation substantially.

In what follows, we provide point-by-point responses to the comments and concerns raised by the reviewer 3.

Sincerely,

Reviewer 3

Reviewer’s Comment: Firstly, I would suggest to structure the manuscript according to the instruction here: https://www.mdpi.com/journal/brainsci/instructions#manuscript. Placing the result section immediately after the introduction makes the understanding of the work difficult. Moreover, I suggest to give a precise definition of the measures called ucd and unit information flow (and sometimes causal flow) before the result section.

Authors’ Response:We apologize for the inconvenience that this matter may have caused the reviewer. Although we used an MDPI journal’s template, we chose the wrong one (i.e., MDPI Journal Entropy) that instructs the authors for having the format we adapted in our initial submission. We addressed this issue in the current version of the manuscript in which the Materials and Methods proceeds the Results and Discussion Sections.

Reviewer’s Comment:Regarding the results in subsection 2.3, in particular, Table 3/4/5 I wonder about the possibility to apply a repeated measure ANOVA for each channel with factor the affect states instead of exploring each pair of affect states.

Authors’ Response: Prior to our analyses, we checked the participants’ unit causal density and the unit causal flow in each of the negative, neutral, and positive affect states (separately as well as combined, with respect to the both individuals and the EEG channels for each of the affect)and found that they did not follow the normal distribution. Therefore, we chose non-parametric tests. We also observed that our analyses of the participants’ unit causal densitiesyielded small effect sizes (Section 3.2. Unit Causal Density, lines 368-374, in the current version of the manuscript).To clarify this, we added the following to the current version of the manuscript (Section2.4.3. Importance of Channels’ Unit Causal Densities, lines 288-302).Therefore, we decided to further apply the bootstrap tests,therebyallowingus to verify the validity of observed significant differences. Specifically, by adapting thistest (i.e., random sampling with replacement) that was carried for 10,000 simulation runs (therefore allowing for potential outliers to be repeated with higher probability and more frequently) at 95% confidence interval (CI) (i.e., p < .05 significance level) enabled us to observe whether these results were indeed reliable, given the small sample in the present study: the CI made it possible to verify thatour results exhibited the central tendency and the overall test showed thatthedistribution of our test results approximated the normal distribution, thereby adhering the law of large number and the central limit theorem. To clarify this, we added the following to the current version of the manuscript (Section2.4.3. Importance of Channels’ Unit Causal Densities, lines338-352).

With regard to our analyses, there are two points that are worth further clarification. They are the choice of non-parametric tests and the follow-up bootstrap test of significance. Prior to our analyses, we checked the participants' unit causal density and the unit causal flow in each of the negative, neutral, and positive affect states (separately as well as combined, with respect to the both individuals and the EEG channels for each of the affect) and found that they did not follow normal distribution. Therefore, we opted for non-parametric analyses. In the case of bootstrap, on the other hand, we realized that our analyses were performed based on a small sample of participants (i.e., fourteen individuals). We also observed that our analyses of the participants’ unit causal flow yielded small effect sizes. Therefore, it was crucial to ensure that any significant results that we observed in our analyses were not due to a subsample of individuals (i.e., distorted data and hence lack of central tendency). This concern was further strengthened by the result of the non-normality of the participants' unit causal density and the unit causal flow. Therefore, we decided to also apply the bootstrap test (i.e., random sampling with replacement) that was carried out for 10,000 simulation runs (therefore allowing for potential outliers to be repeated with higher probability and more frequently) at 95% confidence interval (i.e., p < .05 significance level), thereby enabling us to further verify our results.

However, we can also add the ANOVA in a new Appendix in case the reviewer considers reporting it necessary to further verify our results.

Reviewer’s Comment: In my opinion, the most interesting result is the significant performance of the decoder in the affect state classification but actually authors do not give high relevance to it. Even the title of 2.4 focuses on the channel importance that to me it is like a side result of the classification. I mean firstly I would care about the capability of discriminating between classes and then I would wonder about the feature importance.

Authors’ Response:We fully agree with the reviewer that our manuscript did not discuss the implications of these results adequately. In Section 4. Discussionofthe current version of the manuscript, we addressed the reviewer’s concerns in three steps: 1) relations between our results and the main hypotheses of the affect 2) Correspondence between this relation and the results of the linear model 3) Observations associated with the linear model’s results and future direction. Below, we elaborate these three steps.

Relations between our results and the main hypotheses of the affect: We first explored the relation between our results and the affective workspace [18,23],the bipolarity [19], and the bivalent hypotheses [2022].It reads as follows (lines493-503).

Our findings appeared to be more in line with the affective workspace hypothesis [18,23] than the bipolarity [19] or the bivalent hypotheses [2022]. For instance, the bipolarity hypothesis [19] attributes the positive and negative affect to the opposing ends of a single dimension [24,25]. As a result, it is plausible to expect anti-correlations between these affect states’ flow of information. Contrary to this expectation, we observed positive correlations among them. On the other hand, the bivalent hypothesis [2022] emphasizes on the presence of two distinct and independent brain systems for the positive and negative affect [2022]. This makes it plausible to expect that the information flow associated with these affect states to form non-overlapping and distinct patterns. Although our analyses identified a number of brain regions whose unit causal flows significantly differed between negative, neutral, and positive affect (see Appendix B), we also observed that these regions were common between these affect.”

Correspondence between this relation and the results of the linear model: Next,we discussed the results of the linear model which followed the above paragraph. It reads as follows (lines 504-522).

Further evidence in support of this view came from the classification results of these affect states in which a simple linear model was sufficient to significantly distinguish between them. These results were in line with our correlation analyses in that they implied that any change in the information flow in one affect can be explained in terms of a linear change in the corresponding flow of information in the other affect state that was in the same direction as of the first one. We also observed that a linear model that only utilized the regions with significantly different flow of information, yielded a significantly poorer performance in comparison with the setting in which the whole-brain cortical information flow was considered. Additionally, the mere use of these regions appeared to cause the linear model to treat all three affects as neutral. This interpretation can be verified by observing the sudden and substantial increase in linear model’s correct classification of the neutral affect that was accompanied by its significantly reduced accuracy in the case of both positive and negative affect states. In fact, the recurrence of some of these regions in such domains as social cognition and theory of mind [93], story comprehension [87,88], autobiographical memory [94], decision making [95], working memory [96], and self-referential processing [97], along with the reduced accuracy of the linear model in our study further implied that [1] these regions may not be specific to the affect but may constitute to other cognitive and perceptual events, thereby forming domain-general networks [98,99]. Considering these observations, our findings indicated that the whole-brain flow of information that was manifested in distributed cortical regions were not only shared among the negative, neutral, and positive affect states but also was able to best distinguish between them.

Observations associated with the linear model’s results and future direction:Last, we added a new Section5. Concluding Remarksin which we discussed some of the puzzling parts of the linear model’s results and their implications. We also suggested how the future research may consider to further study these observed effects. It reads as follows (lines 584-610).

On the other hand, a puzzling observation with regards to the linear classifier was the higher percentage of misclassification between the neutral and the negative affect. We noticed that thiscould not readily be attributed to the sample size since we used a balanced dataset in our analyses (i.e., equal number of samples for each of the negative, neutral, and positive affect, perparticipant). Another possible reason behind the observed effect could have been the difference in the shared information between these affect. However, the results of the correlation analyses (i.e., linear measure of mutual information [112]) did not provide any further insight on this matter. For instance, these results could not explain the higher misclassification of the neutral as the negative than the positive affect, despite the fact that the neutral affect appeared to share slightly higher information (i.e., comparably higher correlation) with the positive than the negative affect. Furthermore, the observed correlations between positive and the other two affect states were equivalent. As a result, if the higher shared information was to blame, then it should have caused a comparable misclassification between the positive and the negative which evidently was not the case. Conversely, the lower shared information also was not sufficient to account for the observed higher misclassification of the neutral affect as the negative than the positive affect. This is due to the observation that the high rate of misclassification between the neutral and negative should have in principle been the lowest, given their lowest correlation among the three pairwise comparisons.

The implications of these observations were threefold. First, although the negative, neutral, and positive affect appeared to share a common distributed neural correlates, the underlying dynamics of such a whole-brain cortical information flow might be shared differentially among them. Second, such a dynamics might have more in common between the negative and the neutral affect states than the neutral and the positive or the negative and the positive affect. Third, this potential dynamics that governed the cortical responses to these affect might not primarily be explained through a linear modeling of such a common and distributed neural activity. Therefore, future research that takes into account the non-linear dynamics of neural activity [113,114] in response to negative, neutral, and positive affect is necessary to further extend our understanding of the potential causes of the observed dis/similarities between these affect states.”

Reviewer’s Comment: An overall comment, there are a lot of inaccuracies in the result presentation. Basic information related to the plotted results is missing. Moreover, figure quality should be improved, e.g. axis labels sometimes are repeated even if more plots share the same labels, while there are cases in which labels are written only in the first subplot (see the first column vs the second column of subplots in Figure 1)

Authors’ Response:We apologize the inconvenience that these issues may have caused the reviewer. We updated all the plots and added labels to every subplot within a figure. We also made sure that affect states are capitalized in all the figures (i.e., Negative, Neutral, and Positive instead of negative, neutral, and positive). We have adapted the same convention in all tables.Last, we capitalized all occurrences of these terms that were related to the affect states in the text.

With regards to the information related to each figure, we checked and made sure that their corresponding captions are informative. However, we will also appreciate it if the reviewer can pinpoint any specific types of information that the reviewer is concerned with and we are missing.

Reviewer’s Comment:Introduction line 58: Authors introduced the concept of Granger causality in the framework of the "nonlinear dynamical system analysis". I think this statement should be better explained because the most common implementation of GC is based on the linear AR model, and this is also the case for the Barnett and Seth toolbox. It is not clear to me what type of nonlinearity you are referring to.

Authors’ Response: The reviewer’s concern in fact pinpoint an important pointthat has goneunnoticed by the authors. We modified this part of Section Introduction and added the following to clarify this point (lines 62-76, in the current version of the manuscript).

In this regards, the nonlinear dynamical system analysis [41,42] frames the study of the brain functioning in terms of the interaction between its regions. Specifically, it treats the brain as a complex system [43,44] whose dynamics and ongoing activity [48] orchestrates its cognitive functions [4547,49]. In this respect, the Granger causality (G-causality) [50,51] has found widespread use in neuroscience [5557]. The G-causality is based on predictability and precedence among two or more events that occur at the same time. In the language of G-causality, a variable X is said to G-cause a variable Y if the past of X contains information that helps predict the future of Y over and above information already in the past of Y. Although the Granger causality is based on linear vector autoregressive (VAR) [50,51,56] and hence linear in nature, it approximates [52,53] the transfer entropy [54] i.e., a nonlinear directional measure of mutual information. An advantage of using Granger causality is that unlike the transfer entropy’s complicated estimation [54,68,69], its well-established mathematical formulation and known statistical properties allow for straightforward tests of significance [70,71]. Over the past, there has been concerns for the use of Granger causality in the neuroscience [58,59]. However, a number of subsequent analyses have provided evidence for its utility in such analyses [6062].”

Reviewer’s Comment: - ucd is not defined, it is strange that the next subsection (2.2) titles Unit Causal Density while this measure has been already used before without giving a meaning to it;

Authors’ Response: We apologize for the inconvenience that this matter may have caused the reviewer. Although we used an MDPI journal’s template, we chose the wrong one (i.e., MDPI Journal Entropy) that instructs the authors for having the format we adapted in our initial submission. We addressed this issue in the current version of the manuscript in which the Materials and Methods proceeds the Results and Discussion Sections.

Please also refer to our response to “Reviewer’s Comment:- A formal definition of unit causal density and causal information flow...” for further details.

Reviewer’s Comment:- in Figure 1 part A, B and C are not specified;

Authors’ Response: The subplots were properly labeled (i.e., Figure 2, page 9, in the current version of the manuscript).

Reviewer’s Comment:- it is not clear why 95%CI is [.68,.7] given that the SD=.3;

Authors’ Response: The relation between mean (M), standard deviation (SD), and confidence interval (CI) at 95% can be defined based on the standard error (SE). SE is computed as:

SE = SD/sqrt(N)

and

CI = M +/- 1.96 * SE (where 1.96 is associated with 95% CI)

In our case, N is (Section2.4. Statistical Analysis,lines247-250, in the current version of manuscript) number of channel X number of participants X number of affect (i.e., negative, neutral, and positive) = 62 X 14 X 3 = 2604. Therefore,

X = .3/sqrt(2604) X 1.96 = 0.01152276848

Given M = .69, SD = .3 and N = 2604, we have:

CI = .69 -/+ .0112 = [0.6780.702][0.68 0.70]

Reviewer’s Comment:- 'see A' at the end of the figure explanation, I suppose means Appendix A;

Authors’ Response: As the reviewer mentioned it refers to the Appendix A. We corrected it in the current version of the manuscript.

Reviewer’sComment:- parenthesis line 153;

Authors’ Response: Wecorrected the misplaced parenthesis.

Reviewer’s Comment:- description of Table 4, error in Negative>Neutral

Authors’ Response:Wecorrected this typo in the description of Table 4 (pages15-16,in the current version of the manuscript).

Reviewer’s Comment: - sometimes negative, positive and neutral start with a capital letter other times no, there should be consistency;

Authors’ Response: We capitalized all occurrences of negative, neutral, and positive that were related to affect states in all figures, tables, as well as within the text

Reviewer’s Comment:- figure 4/5/6, according to the colorbar it seems that there are values (blue levels) lower than 0;

Authors’ Response: Weapologize for this misleading information. It was caused by our misalignment of the text after we needed to manually adjust its font size. We have corrected them in all three figures.

Reviewer’s Comment:- figure 7/8/9, figure quality should be improved, i.e. more consistency in the choice of the x and y scale, axis labels not in the last subplot but at the global level;

Authors’ Response:We corrected these issues in the aforementioned figures. We also applied same changes and modifications to all other figures that consisted of subplots.

With regards to Figures 7, 8, and 9, as well as their corresponding tables (i.e., Tables 6, 7, and 8), we moved these figures and their associated content to a new Appendix B Channel-wise Comparison of the Information Flow in Negative, Neutral, and Positive Affect, pages22-27,in the current version of the manuscript. They are Figures Figure A2., Figure A3., and Figure A4.(pages 23-24, in the current version of the manuscript) and Tables Table A2., Table A3., Table A4.,(pages 25-27,in the current version of the manuscript).

The content of this Appendix reads as follows (lines642-655).

Only some of the observed significant differences based on channel-wise paired Wilcoxon rank sum test passed the posthoc two-sample bootstrap test of significance (10,000 simulation runs) at 95.0% confidence interval. In the case of Positive versus Neutral (FigureA2 and Table A2) whereas this test identified CPZ with the higher information flow in Neutral than Positive, it indicated FT7, FT8, P5, P1, P4, P6, PO5, POZ, PO6, and OZ to have higher flow of information in Positive than the Neutral. In the case of Positive versus Negative (Figure A3and Table A3), we observed that whereas Negative affect was associated with higher information flow in F3, Positive affect showed significantly higher information flow than Negative affect in FP1, F7, F2, T8, CP2, P3, P1, P2, P4, PO5, PO8, and OZ. Finally, for Negative versus Neutral (Figure A4and Table A4), the Neutral affect was associated with the higher information flow than Negative in the case of F1, F4, FCZ, C4, and CP2 while Negative affect had higher flow of information than Neutral in F3, FT7, FT8, P5, P4, P6, PO5, POZ, PO6, and O2.

Taken together, the two-sample bootstrap test of significance (10,000 simulation runs) at 95.0% confidence interval identified the relation Positive > Negative > Neutral in channels P4 and PO5 and the relation Positive & Negative > Neutral in channels FT7, FT8, P5, P4, P6, PO5, POZ, and PO6.”

Last, we referred to this Appendix within the main body of the manuscript as follows (page, 14, lines 423-425).

Further details on channel-wise comparison of the information flow in Negative, Neutral, and Positive affect are presented in Appendix B.

Reviewer’s Comment:- table 2/6/7/8 I would suggest to highlight somehow, e.g. bold character, significant channels;

Authors’ Response: We used bold font for the entries of these tables that were significant. We also added the following sentence to their captions.

Bold entry rows indicates the significant difference.

In the current version of the manuscript, Tables 6, 7, and 8 are TablesTable A2., Table A3., Table A4.in Appendix B, pages 23-24.

Reviewer’sComment:- table 6 and figure 7 convey the same type of information (as also the other pairs of table/figure), my opinion is that it is redundant to show both, maybe the histograms can be shown in the appendix;

Authors’ Response:Please refer to our response to Reviewer’s Comment:- figure 7/8/9, figure quality...

Reviewer’s Comment:- table 6, there is a separation between channels where neutral>positive and vice-versa, while in table 7 and 8 not;

Authors’ Response:We corrected for this discrepancy by removing the separation from Table 6 (i.e.,Table A2.., page 25,in the current version of the manuscript).

Reviewer’s Comment: Looking at Table 8, I was surprised noticing that most of the channels are significant while from Figure2C there is no significant difference between negative and neutral. Is there an explanation or am I missing something?

Authors’ ResponseThis is in fact a very interesting observation by the reviewer that adequately hints at what we referred to as emergence of the affect from the variation in the flow of information. Such a difference indeed shows that although these channels were common among the affect and despite the fact that there were no significant difference between negative and neutral at overall unit causal densities (i.e., the overall degree pf causal interactivity), how these unit causal densities were distributed and shared between the channels (i.e., unit causal flows of these channels that expresses the extent to which a channel influenced or was influenced by the remainder of the channels) were significant. In this respect, the linear model also followed a similar trend in which it misclassified the neutral mostly as negative than positive affect.

We added this discussion to Section 5. Concluding Remarks, lines 568-583.

In this article, we provided evidence for the possibility of the emergence of the affect from the variation in the information flow among distributed brain regions that were located in both hemispheres. We supported this viewpoint by showing three results1) the whole-brain cortical flow of information was positively correlated between these affect states, 2)although these distributed regions were shared among the negative, neutral and positive affect, they appeared to share information differentially in response to these affect, 3)a simple linear model was able to distinguish between these affect states with a significantly above average accuracy only when it was provided with the whole-brain cortical flow of information. For instance, whereas we observed a non-significant difference between the channels’ causal densities in the case of Negative and Neutral affect, we found a considerable number of channels’ whose causal flows differed significantly in response to these two affect states. These results hinted at the possibility of interpreting the cortical responses to these affect in terms of the variation in the flow of information that differed significantly among these channels. In other words, the observed differences suggested that although these channels were common between the Negative and the Neutral affect and that their causal densities (i.e., the overall degree of causal interactivity) were non-significant, their causal flows (i.e., the extent of their influence on/by the other channels) were significantly varying in response to these differential affect.

Reviewer’s Comment:- colorbar of figure 10A is not clear, according to the text description values are scaled in [0,1] while in the colorbar negative values are included and the maximum is 0.84. Moreover, the colors are not uniformly distributed.

Authors’ Response:We corrected the colorbar for this figure (i.e., Figure 8 (A), page 18, in the current version of the manuscript). We also changed the confusion matrices associated with the classification results (i.e.,Figure 8 (B)and (C), page 18)with the new one that were generated with Python seaborn for better readability.

Reviewer’s Comment:- Figure 11 "secends"

Authors’ Response: We corrected for this typo in subplot (A) (Figure 1, page 4, in the current version of the manuscript).

Reviewer’s Comment:- A formal definition of unit causal density and causal information flow should be given. The actual description is confusing. I suggest to use a mathematical formula instead of phrasing the concept;

Authors’ Response: We modified the Section 2.3. Causal Density and Causal Flow Computations (pages 5-6, in the current version of the manuscript)to include the missing information. Specifically, we included the mathematical expression for causal density (from Seth [68], page 268).Wealso includedsimplerdefinitions for unit causal density and unit causal flow. The newly added information reads as follows (Section2.3. Causal Density and Causal Flow Computations, lines200-222,in the current version of the manuscript).

Causal density (cd) expresses the overall degree of causal interactivity [68]. It is defined as the mean of all pairwise G-causalities between system elements, conditioned on the remainder of the system. The causal density of a system X (e.g., all the EEG channels in our case) is computed as (Seth [68], page 268):

INSERT EQUATION (1) AT PAGE 6 (in the current version of the manuscript) HERE

and X[ij] denotes the subsystem of X where elements Xi and Xj are excluded. The unit causal density (ucd) of any Xi ∈ X (e.g., a single EEG channel) is then the summed causal interactions that involves Xi, normalized by ∥X∥ i.e., number of elements of X (ibid). This, in turn, results in n ucd values for ∥X∥ = n (in our case, 62 ucd values for 62 EEG channels, per participant, per affect).

On the other hand, the unit causal flow (ucf) of an element Xi ∈ X is defined as the difference between its in-degree and out-degree [68]. In other words, ucf of Xi ∈ X (e.g., ith EEG channel, 1 ≤ i ≤ 62, in our case) expresses the extent to which Xi is influenced by (i.e., in-degree) or influences (i.e., out-degree) the remainder of the elements Xj ∈ X, j ̸= i, 1 ≤ i, j ≤ 62 (e.g., all the other EEG channels in our case).”

Round 2

Reviewer 1 Report

Dear authors, 

This manuscript has been significantly improved.

Nevertheless, it still quite heavy (many figures but especially too many tables, yet not much discussed in the manuscript).

I would recommend a careful reading, but the topic is interesting.

Best regards,

Author Response

First and foremost, the authors would like to take this opportunity to thank the reviewers and the associate editor for their time and kind consideration to review our manuscript. The comments by the reviewers enabled us to improve the quality of our results and their presentation substantially.

In what follows, we provide point-by-point responses to the comments and concerns raised by the reviewers 1.

Sincerely,

Reviewer 1

Reviewer’s Comment: Nevertheless, it still quite heavy (many figures but especially too many tables, yet not much discussed in the manuscript). I would recommend a careful reading, but the topic is interesting.

Authors’ Response: We addressed the concerns raised by the reviewer in two steps: 1) modifying Section 3. Results 2) modifying Section4. Discussion. We explained these changes below.

ModifyingSection 3. Results: Toimprovethe flow of the Section3. Results, we moved Tables 3 through 5 to Appendix B Channel-wise Wilcoxon Test of Significant Difference between the Information Flow in Negative, Neutral, and Positive Affect,pages 20-23, in the current version of the manuscript. We also added the following content to this Appendix.

“TablesA2,A3, and A4summarize the channel-wise paired Wilcoxon rank sum tests on the information flow in the Negative, Neutral, and Positive affect. They are related to Figures 5, 6, and 7, Section 3.3.”

We further updated the captions of these three tables to refer to the appropriate part in the Section3. Results. Specifically, we added the following sentencesto each table:

Table A2., page 21, in the current version of the manuscript:This table corresponds to Figure 5in Section 3.3.

Table A3., page 21, in the current version of the manuscript:This table corresponds to Figure 6in Section 3.3.

Table A4., page 22, in the current version of the manuscript:This table corresponds to Figure 7in Section 3.3.

Subsequently, we updated the references to these tables in Section 3.3. Unit Causal Flowby referring to their Appendix (i.e., AppendixB,in the current version of the manuscript).The updated part of Section3. Resultsthat contains references to these tables reads as follows (lines 414-430, in the current version of the manuscript).

“In the case of Positive versus Neutral, channel-wise paired Wilcoxon rank sum test (Appendix B,Table A2) identified that F5, F4, F6, FC3, C2, C4, CPZ, CP6, and TP8 had significantly higher information flow in the Neutral than Positive states (i.e., Appendix B, Table A2, entries Neutral > Positive). On the other hand, FT7, FT8, TP7, CP3, P5, P1, P4, P6, P8, PO5, POZ, PO6, OZ, and O2 were the channels in the case of Positive affect that had higher flow of information (i.e., Appendix B, Table A2, Positive > Neutral entries). For Positive versus Negative affect states (i.e., Appendix B, Table A3), information flow was higher in the case of Negative than Positive (i.e., Negative > Positive entries) in channels F5, F3, F6, FT7, FC3, FC2, C3, CP3, CPZ, CP6, and O2. On the other hand, Positive affect was associated with higher information flow than the Negative affect (i.e., Appendix B, Table A3, Positive > Negative entries) in channels FP1, FPZ, F7, F1, F2, F4, C4, T8, TP7, CP2, CP4, P7, P5, P3, P1, P2, P4, PO5, PO8, OZ. With respect to Negative versus Neutral (Appendix B, Table A4), we observed that Neutral affect showed significantly higher information flow than Negative affect in channels FPZ, F1, F2, F4, FCZ, C4, T8, and CP2 (Appendix B, Table A4, entries Neutral > Negative). Similarly, the Negative affect wascharacterized with significantly higher information flow in (Appendix B, Table A4, entries Negative > Neutral) F3, FZ, FT7, FT8, CP3, P5, P4, P6, PO5, PO3, POZ, PO4, PO6, CB1, and O2. Further details on channel-wise comparison of the information flow in Negative, Neutral, and Positive affect are presented in Appendix C.”

ModifyingSection4. Discussion: In the previousrevised version of the manuscript (lines 495-505,in the current version of the manuscript) we discussed how the results presented in our manuscript appeared “to be more in line with affective workspace hypothesis [18,23] than the bipolarity [19] or the bivalent hypotheses [2022].(lines 495-496, in the current version of the manuscript).In this respect, the channel-wisecomparison of the informationflow in the Negative,Neutral, and Positiveaffect(i.e.,AppendixC,in current version of the manuscript) were particularly usefulto verify the extent to which our results were not aligned with bivalent hypothesis. We discussedthis matter in the following sentencethat is with reference to the results presented in AppendixC(lines 502-505, in the current version of the manuscript).

Although our analyses identified a number of brain regions whose unit causal flows significantly differed between Negative, Neutral, and Positive affect (see Appendix C), we also observed that these regions were common between these affect.

However, the reviewer’s comment adequately pointed out that we did not properly and specifically discussed the implication of the results associated with the distributed information in our study (lines 505-512). Therefore, we extended the above paragraph by including a short discussion aboutthese findings. We also cited new references [102] through [105] in support of our interpretation in this part. The newly added content reads as follows.

Additionally, the observed flow of information among distributed brain regions that were common between the Negative, Neutral, and Positive affect was also in accord with the affective workspace hypothesis that expresses that the differential affect are the brain states that are supported by flexible than consistently specific set of brain regions [26]. From a broader perspective, our results resonated with the findings that emphasize the importance of the functional connectivity between distributed brain regions that include pre/frontal, parietal, premotor and sensory, and occipitotemporal regions [102] and the implication of such large-scale and distributed networks in the brain functions [103105].

With regards to the previous changes that we applied to our manuscript, we would like to bring to the reviewer’s kind consideration that we further extended the above discussion by connecting its content with the results of the classifier and how these results suggested the whole-brain cortical flow of information resulted in higher accuracy for distinguishing between the Negative, Neutral, and Positive affect (Section Discussion, lines 513-531, in the current version of the manuscript).

In addition, in Section 5. Concluding Remarks, lines 584-592, we briefly discussed how the results of the causal density and causal flow shed light on the potential distributed nature of the brain cortical responses to differential affect states. We followed this (5. Concluding Remarks, lines 593-609)by further discussion on the relation between these results and our observations in the case of correlation analyses and closed this discussion by pointing out (5. Concluding Remarks, lines 610-619) the implications of these observations.

However, we want to emphasize that we are able to apply further changes and/or add further clarifying discussions in case the reviewer believes that there are still missing information that must be included.
